# Exploring the Influence of the Selected Conjugated Fatty Acids Isomers and Cancerous Process on the Fatty Acids Profile of Spleen

**DOI:** 10.3390/cancers16030479

**Published:** 2024-01-23

**Authors:** Tomasz Lepionka, Małgorzata Białek, Marian Czauderna, Wiktoria Wojtak, Ewelina Maculewicz, Agnieszka Białek

**Affiliations:** 1The Biological Threats Identification and Countermeasure Center of the General Karol Kaczkowski Military Institute of Hygiene and Epidemiology, Lubelska 4 St, 24-100 Puławy, Poland; tomasz.lepionka@wihe.pl; 2The Kielanowski Institute of Animal Physiology and Nutrition, Polish Academy of Sciences, Instytucka 3, 05-110 Jabłonna, Poland; m.bialek@ifzz.pl (M.B.); m.czauderna@ifzz.pl (M.C.); w.wojtak@ifzz.pl (W.W.); 3Faculty of Physical Education, Jozef Pilsudski University of Physical Education in Warsaw, Marymoncka 34, 00-968 Warsaw, Poland; ewelina.maculewicz@awf.edu.pl; 4School of Health and Medical Sciences, University of Economics and Human Sciences in Warsaw, Okopowa 59, 01-043 Warsaw, Poland

**Keywords:** cancer, rat spleen, pomegranate seed oil, bitter melon extract, conjugated fatty acids

## Abstract

**Simple Summary:**

The spleen has recently been recognized for its role in lipid metabolism and potential influence on cancer development and progression. Our study investigates effects of dietary supplements, specifically conjugated fatty acids (CFAs) from pomegranate seed oil and bitter melon extract, on concentrations of fatty acids (FAs) of the rat’s spleen in the context of carcinogenesis. Gas-chromatography and silver liquid-chromatography were employed to analyze the concentration of FAs in the spleen. Our research uncovered that supplements added to diets lead to alterations in the concentration of FAs in the spleen of rats, especially fed the diet including the carcinogenic compound (7,12-dimethylbenz[a]anthracene). We observed considerable changes in concentrations of CFAs and disruption in lipid metabolism in the spleen of rats fed the diet including 7,12-dimethylbenz[a]anthracene. These findings underscore the spleen’s vital role in lipid metabolism. Our study suggests that the spleen has a significant impact on cancer progression and treatment.

**Abstract:**

The spleen, traditionally associated with blood filtration and immune surveillance, has recently been recognized for its role in systemic lipid metabolism and potential influence on cancer development and progression. This study investigates effects of dietary supplements, specifically conjugated linolenic acids from pomegranate seed oil and bitter melon extract, on the fatty acid (FA) composition of the spleen in the context of cancerous processes. Advanced methods, including gas chromatography–mass spectrometry and silver ion-impregnated high-performance liquid chromatography, were employed to analyze the spleen’s FA profile. Our research uncovered that dietary supplementation leads to alterations in the spleen’s FA profile, especially under the carcinogenic influence of 7,12-dimethylbenz[a]anthracene. These changes did not align with a simple protective or anti-carcinogenic pattern, as previously suggested in in vitro studies. We observed shifts in conjugated FA isomer concentrations and variations in desaturase activities, suggesting disrupted lipid metabolism in cancerous conditions. The findings underscore the spleen’s vital role in lipid metabolism within the body’s systemic health framework, highlighting the complexity of dietary supplements’ impact on FA profiles in the spleen and their potential implications in cancer progression and treatment. This study adds valuable insight into the complex interplay between diet, disease, and metabolic regulation, particularly in cancerous environments.

## 1. Introduction

Cancer is a complex and devastating group of diseases characterized by uncontrolled cell growth and the ability to invade surrounding tissues [1,2]. As a leading cause of death worldwide, it remains a significant global health challenge, necessitating continued efforts to unravel its complex biology and identify innovative therapeutic approaches [3,4]. Mounting evidence suggests that alterations in lipid metabolism, including fatty acid (FA) composition, contribute to cancer initiation, progression, and therapeutic resistance [5,6,7]. While extensive research has explored various aspects of cancer biology, including the role of FA in cancer cells, understanding the interplay between FA content in surrounding tissues and cancer biology seems crucial.

Traditionally, the spleen has been primarily associated with filtering blood, removing old or damaged red blood cells, and facilitating immune surveillance [8,9,10]. However, recent studies have revealed the spleen’s involvement in diverse functions beyond its classical roles. These include iron metabolism, immune regulation, and an emerging focus on lipid metabolism [11]. Growing evidence suggests that the spleen may influence cancer development and progression through its involvement in systemic lipid metabolism [12]. In this regard, exploring the FA composition of the spleen and its potential modulation by dietary supplements becomes particularly relevant. The spleen, as a highly vascularized organ involved in immune responses and systemic metabolic regulation, may have a significant impact on cancer development and progression through its FA content.

Fatty acids, as essential components of lipids, play critical roles in cellular processes, including energy production, membrane integrity, and signaling pathways [13,14]. Dysregulated FA metabolism has been associated with several diseases, including obesity, diabetes, and cardiovascular disorders. Interestingly, appearing evidence suggests that FA metabolism in cancer cells is substantially altered, allowing cancer cells to adapt to their demanding metabolic needs [14,15,16,17,18]. While much attention has been given to the role of FA in cancer cells, the contribution of FA content in surrounding/peripheral tissues and organs, such as the spleen, to the cancerous process remains largely unexplored.

Conjugated linolenic acids (CLnAs) are a subgroup of 18-carbon unsaturated FAs with unique structures (system of conjugated double bonds) called conjugated fatty acids (CFA) and potential health benefits, naturally present in seeds of various plants, including edible, like pomegranate (*Punica granatum*, Lythraceae) and bitter melon (*Momordica charantia*, Cucurbitaceae). Those natural CLnAs sources, rich in polyphenols, FA, vitamins, and minerals (pomegranate) or cucurbitane type triterpenoids and glycosides, phenolic acids, flavonoids, carotenoids, essential oils, sterols, saponins, amino acids, some proteins as well as micro and macroelements and vitamins (bitter melon), have been both consumed and traditionally used for their medicinal properties [19,20,21,22,23]. CLnA isomers, especially punicic acid (*cis*-9, *trans*-11, *cis*-13 C18:3, c9t11c13C18:3, PA) found in pomegranate and α-eleostearic acid (*cis*-9, *trans*-11, *trans*-13 C18:3, c9t11t13C18:3, ESA) present in bitter melon, are considered as bioactive FA. Recent studies have demonstrated the anticancer potential of CLnAs, including their ability to inhibit tumor cell growth, induce apoptosis, and modulate key signaling pathways [24,25]. However, the impact of dietary supplementation with CLnAs from *Punica granatum* and *Momordica charantia* on the FA composition of the spleen and its implications for cancerous processes are still to be investigated.

This study aims to bridge the existing knowledge gap by quantitatively analyzing the FA content in the rats’ spleens and characterizing its composition in terms of the cancerous process. Employment of state-of-the-art analytical techniques, such as capillary gas chromatography–mass spectrometry (GC-MS) and silver ion-impregnated high-performance liquid chromatography with photodiode array detection (Ag^+^-HPLC-DAD), allowed optimal identification and quantification of individual FA in spleen tissue samples from animals supplemented with CLnAs. The applied Ag^+^-HPLC-DAD method for identification and quantification of CFA makes a good complement to the most commonly used GC-MS methods as it does not need preliminary derivatization into volatile compounds and ensures the absence of artifacts. Thanks to the use of four analytical columns imprinted in Ag^+^, connected linearly, it enables the analysis of the entire CFA profile [26]. The findings of this study could provide valuable insights into the interplay between FA in the spleen and cancer etiology and biology, with a specific focus on the influence of dietary supplements containing CLnAs. Understanding the potential modulation of the spleen’s FA content by pomegranate- and bitter-melon-derived supplements may reveal novel mechanisms underlying their anticancer effects.

## 2. Materials and Methods

### 2.1. Dietary Supplements

Cold-pressed, unrefined oil extracted from seeds of pomegranate fruits (pomegranate seed oil, PSO), originating from Great Britain (100%, ECOSPA), was purchased from the local market in Warsaw, Poland. It was protected from light in a dark glass bottle, and it was stored unopened at 8 °C in the original manufacturer’s package. After bringing it to ambient temperature, it was administered to animals via gavage, in the amount of 0.15 mL per animal daily. Bitter melon tea for brewing (Tra Kho Qua, Hung Phat Corp, Vietnam), consisting of bitter melon dried fruits, was purchased from the local market in Warsaw, Poland. Bitter melon fruit aqueous extract of 1% (*w*/*v*) (BME) was prepared fresh daily. Hot water (80 °C) was added to the weighted portion of the dried material, left for 10 min, and filtered. After bringing it to ambient temperature, fresh BME was administered to animals as drinking fluid daily ad libitum. Detailed characterization and content of bioactive compounds in both applied botanicals were given previously. PSO contained 26 FAs, of which PA, ESA, c9c12 C18:2 linoleic (LA), c9C18:1 oleic (OL), C16:0 palmitic, and C18:0 stearic acids were present in the highest amounts. BME contained 10 FAs in rather small amounts and among them, C16:0 and C18:0 were present in the highest amounts. In PSO administrated to animals, only conjugated trienes (CTs) were detected, of which ttt CT isomers predominated, followed by cct CT isomers and ttc CT isomers, whose share was the smallest. The presence of PA and ESA was confirmed. BME contained a relatively small amount of CFAs, among which CTs predominated [27].

### 2.2. Animal Experiment

The animal experiment was carried out after obtaining the 2nd Local Ethical Committee on Animal Experiments consent (No. 56/2013 and 54/2015) following European Union Directive 2010/63/EU for animal experiments. The detailed design of the animal experiment was thoroughly described previously [27]. Sprague–Dawley rats (females, n = 96, age 30 days) were purchased from the Central Laboratory of Experimental Animals, Medical University of Warsaw (Warsaw, Poland). During the entire experiment, animals were housed in an animal room at a constant temperature of 21 ± 1 °C with a 12 h light–dark cycle and relative humidity of 50–60% in plastic cages (2 individuals per cage). They were fed the standard laboratory chow Labofeed H ad libitum (Feed and Concentrates Production Plant, A. Morawski, Żurawia 19, Kcynia, Poland) and had unrestricted access to fresh drinking water or BME. The detailed composition of Labofeed H (per kg of diet) was previously published. The detailed analysis of Labofeed H showed that there were 18 FAs present in the feed, of which LA, c9c12c15 C18:3 α-linolenic (ALA), OL, C16:0, and C18:0 acids were predominant. The examined fodder did not contain any CFA [27]. After a 1-week adaptation period, animals were randomly divided into the following eight groups (12 animals per group):-CON and CONplus—control groups without diet supplementation, fed a standard diet and water ad libitum,-M and Mplus—animals fed a standard diet supplemented with 1% aqueous extract of bitter melon dried fruits (BME) ad libitum,-G and Gplus—animals were fed the standard diet and water ad libitum and were given 0.15 mL/d PSO via intragastric gavage,-GM and GMplus—animals were fed the standard diet and were supplemented with both 0.15 mL/d PSO via intragastric gavage and 1% BME ad libitum.

At the 37th day of life, diet supplementation started, which lasted for 21 subsequent weeks. Daily fodder and drinking fluid intake did not differ among the control and experimental groups. For the induction of mammary tumors, rats from four experimental groups defined as “plus” were given 7,12-dimethylbenz[a]anthracene (DMBA) at the dose of 80 mg/kg body weight on the 50th day of life. DMBA was administered as a solution intragastrically via gavage after dissolving directly in PSO (in the case of Gplus and GMplus groups) or in rapeseed oil (in the case of CONplus and Mplus groups). During the whole experiment, animals were also monitored daily for any specific signs of welfare disorders (e.g., appetite loss, ruffling, sluggishness, apathy, hiding, curling up). They were also checked for specific signs of health deterioration and weighed weekly. There were no differences in feed and liquid intake between experimental groups, which was checked during the entire experiment. Main growth parameters as well as masses of internal organs were presented previously [27]. After the experimental period, all animals from each group were decapitated and exsanguinated. Spleens were extracted, washed with saline solution, and kept deep frozen (−70 °C).

### 2.3. Fatty Acids Analysis by Gas Chromatography Coupled with Mass Spectrometry (GC-MS)

Prior to chromatographic analyzes, spleen samples (~50 mg) were subjected to alkaline hydrolysis. They were treated with a mixture of KOH in water and KOH in methanol solutions. Next, both internal standard solutions (nonadecanoic acid and sorbic acid) were added. The mixture was flushed with argon (Ar), vortexed, and sonicated under the stream of Ar at 95 °C for 10 min. The obtained mixture was protected from the light and stored in the sealed vial at ~22 °C overnight. Next, water was added to the hydrolysate and the solution was acidified with 4 M HCl to ~pH 2. Free fatty acids were then extracted with methylene chloride (2 × 3 mL) and n-hexane (2 × 3 mL). Both organic layers were combined, dehydrated with anhydrous Na_2_SO_4_, and removed under the stream of Ar. The obtained residue was dissolved in 1.0 mL of n-hexane.

FA content of examined spleen samples was determined derivatized as fatty acid methyl esters (FAMEs) after the base- and acid-catalyzed methylations procedure. Next, 0.5 mL of hexane solution was heated in 40 °C for 60 min with 2M NaOH solution in methanol. After cooling to 5–10 °C sample was heated at 40 °C for 60 min with 50% BF_3_ solution in methanol. A total of 2.5 mL of supersaturated NaCl solution was added to the cooled sample, and FAMEs were extracted with 2.5 mL of n-hexane. FAMEs were determined by capillary gas chromatography coupled with mass spectrometry (GC-MS) using the gas chromatograph (Shimadzu GC-MS-QP2010 Plus EI, Tokyo, Japan) coupled with quadruple mass selective detector (Model 5973N, Shimadzu, Tokyo, Japan). Moreover, the chromatographic system was equipped with a BPX70 fused silica column (120 m × 0.25 mm i.d. × 0.25 μm film thickness; Phenomenex, Torrance, CA, USA) and an injection port [28]. Nonadecanoic acid (99%, Sigma, St. Louis, MO, USA) was used as the internal standard (IS).

FAMEs identification was based on electron impact ionization spectra of FAMEs and compared to authentic FAMEs standards: Supelco 37 Component FAME Mix, nonadecanoic acid (C19:0; as the internal standard in GC-MS analyses) (Sigma, St. Louis, MO, USA), Bacterial Acid Methyl Ester (BAME) Mix (Sigma, St. Louis, MO, USA), c9,t11C18:2—RA methyl ester standard (Sigma, St. Louis, MO, USA), t10,c12C18:2 (Sigma, St. Louis, MO, USA), c9t11c13C18:3—PA methyl ester standard methyl punicate (Matreya LCC, State College, PA, USA), c9t11t13C18:3—α-eleostearic ESA alpha methyl ester standard (Larodan Fine Chemicals, Solna, Sweden), and the NIST 2007 reference mass spectra library (National Institute of Standard and Technology, Gaithersburg, MD, USA). All FAME analyses were based on total ion current chromatograms (TIC) and/or selected-ion monitoring chromatograms (SIM) (Appendix A). Three parallel samples were prepared from each spleen sample. Results are expressed as μg/g of fresh tissue (or mg/g of tissue). FA profile was expressed as % of the total FAs pool comparing the content of individual FA to the total FAs content.

### 2.4. Determination of CFA Content Using Ag^+^-HPLC-DAD

CFAs in examined spleen samples were analyzed as free CFAs, without any derivatization, with argentometric high-performance liquid chromatography with photodiode array detection (Ag^+^-HPLC-DAD). Prior to chromatographic analyses, samples of spleen were subjected to alkaline hydrolysis according to the method of Czauderna et al. [29,30]. The obtained residue was dissolved in 0.5 mL of n-hexane, vigorously vortexed, and transferred into the vial and then 10 μL of the resulting solution was injected into the Ag^+^-HPLC-DAD system. A Waters HPLC 625LC system (Waters, Milford, MA, USA) equipped with a photodiode array detector (DAD) (Waters, Milford, MA, USA) operated in a UV range from 195 to 400 nm was used for the detection of CFA isomers. Four analytical ion-exchange columns loaded with silver ions (Chrompack ChromSpher, 5 μm, Lipids, 250 mm × 4.6 mm; Varian, Middelburg, the Netherlands) were used in conjunction with a guard column of 10 × 3 mm containing the same stationary phase. Sorbic acid (the conjugated FA) was used as the internal standard (^CFs^IS) [30].

The ambient temperature was 22–24 °C, while a column heater maintained the temperature at 23 °C. Ag^+^-HPLC-DAD system pressure was 15.25 ± 0.08 MPa. The samples were subjected to isocratic elution (2 mL/min) using a mobile phase composed of n-hexane, glacial acetic acid, and acetonitrile (98.4:1.6:0.0125, *v*/*v*/*v*). The columns were equilibrated with freshly prepared mobile phase at least 35 min before sample injections. The mobile phase was carefully stirred before chromatographic analysis as the reproducibility of the fractionation was sensitive to small fluctuations in the concentration of acetic acid and more to the concentration of acetonitrile. Detection of CFAs was conducted at 234 nm for CFAs containing two conjugated double bonds (conjugated dienes—CDs) and at 270 nm for CFAs with three conjugated double bonds (conjugated trienes—CTs). ^CFs^IS was monitored at 259 nm [30]. Identification of CFAs as CDs or CTs was based on retention times and UV spectra of analytical standards of CFAs (i.e., CLA isomer mixture; c9t11C18:2; t10c12C18:2; CLnA isomers: t8t10c12C18:3—α-calendic; t9t11c13C18:3—catalpic (CA); c9t11t13C18:3—α-eleostearic (ESA alpha); t9t11t13C18:3—β-eleostearic (ESA beta); and c9t11c13C18:3—punicic acid (PA) (Larodan Fine Chemicals, Solna, Sweden)). Sorbic acid (c2c4C6:2; as ^CFs^IS) was supplied by Sigma (St. Louis, MO, USA). All results were expressed as μg/g of fresh tissue or as CFA profile (% of CFAs).

### 2.5. Calculation of Indices in Spleen Samples

Based on the FA content, determined in the spleen samples, the following indices were calculated [31]:A-SFA = C12:0 + C14:0 + C16:0; 
T-SFA = C14:0 + C16:0 + C18:0; 
_index_A^SFA^ = (C12:0 + 4 × C14:0 + C16:0)/(ΣMUFA + Σn-6PUFA + Σn-3PUFA); 
_index_T^SFA^ = (C14:0 + C16:0 + C18:0)/[(0.5 × ΣMUFA + 0.5 × Σn-6PUFA + 3 × Σn-3PUFA)/Σn-6PUFA)]; 
^C18:0^∆9_index_ = c9C18:1/(c9C18:1 + C18:0); 
^Σ∆9,6,5,4^FA_index_ = (ΣMUFA + ΣPUFA)/(C16:0 + C18:0 + C20:0 + C22:0 + C24:0 + ΣMUFA + ΣPUFA); 
^n6ElongC20/C18^index = c11c14C20:2/(c11c14C20:2 + c9c12C18:2); 
^n3ElongC22/C20^index = c7c10c13c16c19C22:5/(c7c10c13c16c19C22:5 + c5c8c11c14c17C20:5);
Δ4_index_ = c4c7c10c13c16c19C22:6/(c4c7c10c13c16c19C22:6 + c7c10c13c16c19C22:5); 
Δ5_index_ = c5c8c11c14C20:4/(c5c8c11c14C20:4 + c8c11c14C20:3);
h/H-Ch = (c7C18:1 + c9C18:1 + c12C18:1 + c14C18:1 + c11C20:1 + 13C22:1 + c9c12C18:2 + c9c12c15C18:3 + c6c9c12C18:3 + c5c8c11c14C20:4 + c11c14C20:2 + c5c8c11c14c17C20:5 + c7c10c13c16C22:4 + c7c10c13c16c19C22:5)/(C14:0 + C16:0)

### 2.6. Statistical Analysis

All data were presented as mean values ± standard deviation. For variables with skew distribution, data were transformed into logarithms, retransformed after calculations, and presented as mean and confidence interval. Statistica 13.5 software (StatSoft, Cracow, Poland) was used for the statistical analysis. For variables with normal distribution obtained data were tested with one-way ANOVA and post hoc Tukey test (marked * in tables). For variables without normal distribution, data were tested with the Kruskal–Wallis test, which is a non-parametric equivalent of one-way ANOVA, with post hoc Dunn’s test. For values of indices calculated based on FA content, the Kruskal–Wallis test with post hoc Dunn’s test were performed. The acceptable level of significance was established at *p* < 0.05.

To verify whether the diet modifications and applied experimental conditions significantly affected group diversity, chemometric analyses were performed using Statistica 13.5 software. FAs and CFAs content in spleen samples were used as descriptors to study a possible discrimination of the spleen samples. Prior to analyses, the original data were transformed into natural logarithms and then auto-scaled (standardized).

Cluster analysis was carried out using the agglomeration method. Euclidean distance was used as the distance determination method, and the Ward method was used as the agglomeration method. The application of the more restrictive Sneath’s criterion (33%) was used for dendrograms’ analysis and cluster distinguishing. To determine the differences among existing clusters of spleen samples, one-way ANOVA with post hoc Tukey test was performer. The accepted significance level was established at *p* < 0.05. Similarity analysis was performed by grouping features and objects for variables differing significantly among existing clusters to prepare a heat map.

## 3. Results

### 3.1. FA Profile of Rats’ Spleen

GC-MS analysis of spleen samples enabled the determination of 24 FAs including 7 saturated fatty acids (SFAs), 8 monounsaturated fatty acids (MUFAs), and 9 polyunsaturated fatty acids (PUFAs) (Table 1, Figure 1). Predominating FAs were C16:0 and C18:8 from SFA, c9C18:1 from MUFAs, and c9c12C18:2 and c5c8c11c14C20:4 from PUFAs. Experimental conditions (diet supplementation and DMBA treatment) influenced FAs, as the content of almost all quantified FAs (except c6C18:1) differed significantly among experimental groups. The highest content of C16:0 was determined in the GM group, which significantly exceeded the content in all groups treated with DMBA, whereas the lowest content of C16:0 was detected in the CONplus group. The spleen of animals from the GM group was also the most abundant in C18:0, of which high content was also detected in M group. Likewise, the GM group exhibited the highest overall SFA content, significantly surpassing the values observed in groups treated with DMBA. Moreover, a noticeable trend toward decreasing SFA content was observed in groups exposed to the carcinogen.

In terms of MUFAs, the GM group exhibited the highest content of c9C17:1 and c9C18:1, significantly exceeding the values observed in the G and GMplus groups, as well as the M and Mplus groups, respectively. Additionally, GM displayed the highest content of ΣC16:1 and c11C18:1, with significant differences observed compared to their content in the CONplus group (c11C18:1) or the G, CONplus, and Mplus groups (ΣC16:1). The c6C18:1 and c7C18:1 were exclusively absent in the G group, although the content of these FAs was similar across the remaining groups, except for a significant variation between GM and Mplus groups. Moreover, significant differences were observed in the overall MUFA content among the groups, with the GM group having the highest content and the CONplus group displaying the lowest content, significantly different from the CON and GM groups.

In terms of PUFAs’ content, significant variations among different groups were observed. The GM group exhibited the highest content of seven out of the nine detected PUFAs. Specifically, the highest content of c9c12C18:2 was found in the GM group, which was significantly higher compared to all “plus” groups. On the other hand, the CONplus group displayed the lowest content of this FA, and it was significantly lower compared to all groups not exposed to DMBA. Similar trends were observed for c11c14C20:2 and c7c10c13c16c19C22:5, with the CONplus group having the lowest content of these PUFAs compared to the animals not treated with the carcinogen. Additionally, the highest content of c9t11C18:2 was determined in the GM group, significantly surpassing the values observed in the “plus” groups, except for GMplus. Notably, c9t11C18:2 was not detected in the CON and M groups. Regarding c8c11c14C20:3, the groups receiving supplementation (M, G, GM) exhibited the highest levels of this FA, which were significantly higher compared to the remaining groups. Conversely, Gplus and Mplus groups had the lowest content of c5c8c11c14C20:4, which was significantly lower compared to the G, M, and GM groups. Similarly, the c5c8c11c14c17C20:5 content was the lowest in the Gplus group, and it was significantly lower compared to the M, G, GM, and CONplus groups. Furthermore, the overall content of ΣPUFA and ΣFA (total FA) showed significant differences among the groups, with the GM group displaying the highest content. Significant differences were observed among all “plus” groups. Notably, there was an apparent trend towards decreasing ΣPUFA and ΣFA content in groups exposed to the carcinogen, particularly in the CONplus group. This group exhibited significantly lower levels of these FAs compared to the groups not exposed to DMBA.

The FAs’ percentage share in the material varied among different groups (Table 2). In terms of individual FA, it was observed that the G group had a significantly lower share of C14:0 compared to the M, CONplus, and GMplus groups. For C15:0, the GMplus had the highest percentage share, significantly higher compared to all groups not exposed to the DMBA and Mplus groups. Similarly high levels of this FA were determined in CONplus group. C16:0 was one of the most prevalent FAs in the pool, yet significant differences in its percentage share were observed only between the CONplus and GMplus group, which was characterized by the highest levels of this FA. In terms of iso-C17:0 (i-C17:0), the spleen of the GM group exhibited significantly lower levels compared to the Mplus and GMplus groups. For C20:0, the CONplus group had significantly higher levels compared to the G and Gplus groups supplemented only with PSO. In terms of ΣSFA share in the FA pool, it ranged from 36.97 ± 1.82% to 39.27 ± 1.16%, making it the second largest FAs group behind ΣPUFA. The lowest ΣSFA share was determined in the CON group, in comparison especially to GMplus group, which showed the highest share of ΣSFA among all groups.

The percentage share of certain MUFA in the spleens varied among groups. The share of ΣC16:1, c7C18:1, and c9C18:1 remained relatively consistent across the different groups, while no significant differences were observed in their levels among the groups. The levels of c11C18:1 showed significant variations among groups, with the lowest proportion observed in both the G and GM groups, significantly lower than in the spleen of “plus” groups. For c11C20:1, the CONplus group had the highest percentage, significantly higher than most groups not exposed to DMBA, except the M group. When considering MUFA in total, significant differences were observed between the G and Mplus groups, with the latter characterized by higher levels of ΣMUFA.

ΣPUFA were the most prevalent group, comprising a range of 44.16 ± 6.40% to 49.51 ± 1.57% of the total FA pool. Significant differences were observed in the percentage share of c9c12C18:2 among the groups. CONplus had significantly lower levels compared to CON, G, GM, and Gplus. The G group exhibited the highest abundance of this FA, with significantly higher levels compared to M, CONplus, and Mplus. Regarding c9c12c15C18:3, both CON and CONplus groups had significantly higher levels compared to the remaining “plus” groups, with no differences observed among groups not exposed to DMBA. No significant differences were found in the share of c9t11C18:2 and c4c7c10c13c16c19C22:6. The levels of c11c14C20:2 were significantly higher in both groups supplemented with PSO (G and Gplus) compared to the CONplus group, which exhibited the lowest level of this FA.

For c5c8c11c14C20:4, the highest share was observed in the CONplus group, significantly higher compared to the CON and GM groups. Similarly, the CONplus group exhibited the highest levels of c5c8c11c14c17C20:5, which were also significantly higher than in the CON and GM groups, as well as in the Gplus and GMplus groups. The highest share of c7c10c13c16c19C22:5 was observed in the M and G groups, significantly differing from the “plus” groups. In terms of ΣPUFA, the G group had the highest share, significantly higher than the “plus” groups except for CONplus. Moreover, no significant differences were observed within the particular groups of non-exposed and DMBA-exposed subjects.

The ΣSFA levels showed significant differences among the groups (Table 3). The CON and M groups had higher ΣSFA levels compared to the CONplus and Mplus groups. The G group had elevated ΣSFA levels compared to the CONplus group. The GM group had higher ΣSFA levels compared to the CONplus, Mplus, Gplus, and GMplus groups. The proportion of ΣMUFA also exhibited significant differences among the groups, as GM and CON groups were characterized by their highest levels and were statistically higher compared to CONplus (both GM and CON) and Mplus (GM only).

The ΣPUFA content was higher in groups not exposed to DMBA. The highest levels of PUFA were detected in the GM group and were significantly higher compared to all “plus” groups. The lowest proportion of ΣPUFA was detected in the CONplus group, and it was significantly lower compared to all groups not exposed to DMBA. Similar observations were made for Σn6 PUFA. Regarding Σn-3PUFA, M and GM groups were most abundant in those FA and had significantly higher levels compared to all “plus” groups. Interestingly, the lowest levels of Σn-3 PUFA were detected in groups both supplemented with PSO and/or BME and exposed to DMBA (Mplus, Gplus, GMplus), which had significantly lower contents compared to all groups not exposed to the carcinogen. Likewise, the ratio of Σn-3PUFA to Σn-6PUFA (Σn-3PUFA/Σn-6PUFA) was lowest in Gplus and GMplus groups, and the values were significantly lower compared to groups M, G, CONplus, and CON (GMplus only). The highest value of this ratio was detected in the M group.

The level of long-chain polyunsaturated fatty acids (ΣLPUFA) was highest in the GM group, and it was significantly higher compared to CON and all “plus” groups. On the other hand, the spleen of rats from Mplus and Gplus groups were characterized by the lowest levels of these FAs and were significantly different to their respective groups not exposed to DMBA (M and G). Similar observations apply to Σn-3PUFA and Σn-6PUFA, as their highest levels are detected in the GM group, and their levels in remaining supplemented groups (M and G) are significantly higher compared to their corresponding groups exposed to DMBA (Mplus, Gplus, GMplus). Interestingly, no differences were observed between the CON and CONplus groups.

As far as atherogenic saturated FAs (A-SFA) are concerned, their highest levels were detected in the GM group (significantly higher compared to all “plus” groups), and the lowest were detected in the CONplus and Mplus groups. In terms of the A-SFA/ΣFA ratio, no significant differences were observed. For the ratios of ΣSFA to unsaturated fatty acids (UFA) (ΣSFA/ΣUFA) and ΣSFA to ΣPUFA (ΣSFA/ΣPUFA), significant differences were observed among the groups (*p* < 0.05). Group G had significantly lower values compared to groups CON, M, Gplus, and GMplus for ΣSFA/ΣUFA, while group G had a significantly higher ratio compared to groups CON, M, and GM for ΣSFA/ΣPUFA. For the ratios of ΣSFA to ΣMUFA, there is a significant difference between the G and Mplus groups, with the lowest value observed in the Mplus group. As the ratio of ΣSFA/ΣFA is concerned, the value in the CON group was significantly lower compared to the CONplus and GMplus groups. Any significant differences between groups in values of ΣMUFA/ΣFA ratio, ^C18:0^Δ9_index_, ^ΣΔ9,6,5,4^FA_index_, and ^n6ElongC20/C18^index were observed. Conversely, ^n3ElongC22/C20^index values differed significantly between GM and CONplus groups. For the Δ4_index_, in most of the groups, significant differences were exhibited, with groups of healthy animals (M, G, GM) showing meaningfully lower values of the indices compared to all groups exposed to DMBA. Similar observations were made for Δ5_index_, as its values in groups M and G were significantly lower compared to CONplus and Mplus groups. In the case of the ratio of hypo- to hyper-cholesterolemic FAs (h/H-Ch), a significant increase in the GM group compared to the GMplus group was observed.

Detailed characteristics of CFA content in spleen samples are presented in Table 4. As far as total CFA content is concerned, the CON group exhibited significantly higher content compared to the M, G, and Mplus groups, while the Mplus group showed the lowest level of these Fas. Similarly, the CD content was significantly higher in the CON group compared to the M and Gplus groups. As far as tt isomers content is concerned, Gplus showed their highest values compared to the M and G groups. Regarding ct/tc isomers, the CON group contained their higher levels compared to the M, CONplus, Mplus, and GMplus groups. Interestingly, the only significant difference noticed in cc isomer content was observed among groups supplemented only with BME, with the Mplus group showing a higher cc content than the M group.

There is a pronounced variability in CT content across all groups, indicating a robust dependency of CT levels on experimental conditions. The highest levels of CT isomers were observed in the CON and GMplus groups, while the M group had the lowest content of these isomers among all groups. Similarly, the content of ttt isomers varies greatly among animal groups, with the CON and GM plus groups showing higher levels and the M group having considerably lower levels. Also, regarding ttc/ctt contents, the M group presented significantly lower levels compared to the CON, CONplus, Gplus, and GMplus groups. The cct isomers content was also highly varied, with groups like Gplus and CON showing markedly higher levels of these Fas, while the G group presented the lowest levels of these compounds of all groups.

The CFA content expressed as the percentage share of individual groups (Table 5) was also analyzed. The CD isomer share was significantly higher in the M group in comparison to the remaining groups, especially CON and GM and GMplus. Similar observations apply to the tt isomers percentage share in the M group compared to the CON and GM groups. Regarding the ct isomers share, it was the highest in the G group compared to almost all “plus” groups. Generally, exhibition to DMBA seemed to decrease ct isomers share, but mostly in the GMplus group. Different observations were made in the case of cc isomers share, which was the highest in the Mplus group and was significantly different from its share in the CON and M groups. Similar dependencies were observed in the case of the share of ttt isomers, which were the most abundant in CTs group. Regarding ttc/ctt isomers, significant differences between Gplus and M groups were observed, with the highest proportion of these isomers in the Gplus group. The cct isomers share was the lowest in the G group, which was significantly lower compared to CON, GM, Mplus, and Gplus.

The share of particular isomers within the CDs and CTs groups was also analyzed. The lowest proportion of tt isomers in the total CD content was observed in the CON group, and the highest was observed in CONplus and GMplus. On the other hand, ct isomers percentage share was the highest in the CON group, and it was significantly higher compared to the CONplus, Mplus, and Gplus groups. Regarding the percentage share of cc isomers, it was the highest in the Mplus group, while lower levels were observed in the CON, M, and G groups. As far as particular isomers of CTs are concerned, the most abundant were ttt isomers, and there were significant differences observed among groups. The largest share of these CT isomers was observed in the CON and GMplus groups, while the lowest was observed in the M group. On the contrary, the M group showed the largest share of ttc/ctt isomers, which was significantly higher compared to CON and GMplus. Also, in the M group, the most abundant were cct isomers, whose share in this group was significantly higher compared to all remaining groups.

### 3.2. Cluster Analysis (CA)

The results of CA are presented as dendrograms in Figure 2 and Figure 3. The application of the more restrictive Sneath’s criterion (33%) to the dendrogram analysis allowed us to distinguish of three clusters, which differentiate the examined variables (Figure 2). The first cluster (Cl1) included: C14:0, C15:0, ΣC16:1, c9C18:1, c11C18:1, c9c12c15C18:3, c9C17:1, C22:0, C20:0, c11C20:1, and c5c8c11c14c17C20:5; the second cluster (Cl2) included ten variables: C16:0, C18:0, c9c12C18:2, c11c14C20:2, c8c11c14C20:3, c5c8c11c14C20:4, c7c10c13c16c19C22:5, c4c7c10c13c16c19C22:6, i-C17:0, and FA total; and the third cluster (Cl3) included all eight variables describing CFA (CFA, CD, CT, tt, ct, cc, ttt, ttc/ctt, cct). The application of the more restrictive Sneath’s criterion (33%) to the second dendrogram analysis allowed us to distinguish four clusters, which differentiated the examined spleen samples (Figure 3). The first cluster (Cl1) included individuals: GM1, GM3, GM7, GM8, GM10, and M4; the second cluster (Cl2) included mostly individuals from DMBA-treated groups not supplemented with PSO; the third cluster (Cl3) included most of the spleen samples obtained from animals treated with DMBA and supplemented with PSO; whereas the fourth cluster (Cl4) included most samples of spleen obtained from animals non-treated with DMBA treatment. Similarity analysis was performed by a grouping of features and objects to prepare a heat map (Figure 4). Statistical analysis revealed significant differences in the content of all examined variables among existing clusters of spleen samples (Table 6).

## 4. Discussion

The spleen is the largest lymphoid organ in the body, dark red to blue-black, and constitutes an important part of the reticuloendothelial system (RES), also known as the mononuclear phagocytic system. It is located in the abdomen just below the diaphragm. Its functions include phagocytosis of damaged or old cells and particulate matter, recycling of iron, involvement in immune responses, interception of blood components, and erythropoiesis [8,10,32]. Its structure is unique and corresponds to its function by being compartmentalized into different regions with adaptations not found in other lymphoid organs. The white matter, which acts as an immune organ, is the site of production and maturation of B and T cells. It regulates the response to inflammation and infection. It is composed of lymphoid tissue and consists of a periarterial lymphatic sheath and germinal center. Acting as a phagocytic organ, the red pulp filters the blood, removing old, damaged, or defective red blood cells (RBCs), antibody-coated blood cells, antibody-coated bacteria, and serves as a reservoir of blood components, especially white blood cells (WBCs) and platelets. It also harvests iron from old RBCs for recycling into new RBCs and can produce new RBCs if the bone marrow is not functioning properly. The red pulp is composed of macrophages and granulocytes, which cover the vascular space (cordate and sinusoidal). Spleen macrophages, along with liver macrophages, are reservoirs of iron derived from erythrocyte degradation and can release iron in response to stimuli from the bone marrow. The spleen is also the site where antibody production to many antigens is organized through the cooperation of T lymphocytes, antigen-presenting cells, and B lymphocytes. The spleen is located in the left hypochondrium and is surrounded by the peritoneum except for the pulmonary hilum. It is perfused by two different systems: closed and open circulation. The majority of blood (over 90%) flows directly from the capillaries into the venous sinus and is not filtered by the red pulp. A small amount of blood flows from the capillaries into the Billroth’s cords and slowly perfuses the spleen through an open circuit that enters the venous sinus through a discontinuity in the endothelial basement membrane [9,32,33,34]. In the spleen of rodents, some of the smallest arterial branches terminate in the marginal sinus whereas others traverse the marginal zone to form the venous system of the red pulp, which makes it slightly different on an anatomic level than in humans [8].

The spleen is an important organ for immune homeostasis as both innate and adaptive immune responses can be efficiently mounted in the spleen [8]. A dysfunctional spleen has been associated with the development of a variety of diseases, including sickle cell anemia and malaria. Splenectomy (removal of the spleen) may be necessary to prevent fatal bleeding following injury, to treat diseases that destroy blood cells, or to treat cancers involving the spleen. Metastasis to the spleen is very rare and occurs with multi-organ metastasis or dissemination in the late stages of cancer. Breast, lung, ovarian, colorectal, renal, and skin cancers are the most frequent primary cancers metastasizing to the spleen [35]. Splenomegaly is considered to be connected with poor prognosis in different types of leukemia [36]. Splenomegaly that developed after abdominal radiotherapy for gastric cancer later resulted in sepsis, pneumonia, prolonged hospitalization, sepsis-related death, and increased economic burden for the patient [37]. It was also observed that splenectomy directly correlated with a significant increase in malignancy induction, as splenectomized rats and mice showed a significant increase in malignancy induction [38]. In an epidemiological study by Kristinsson et al. increased risk of not only hematologic malignancies in splenectomized patients but also many solid tumors (lungs, cheek, esophagus, pancreas, liver, colon, and prostate) occur/was reported [39]. In the pancreatic cancer animal model, the study animals recorded an increase in peritoneal metastasis and progressive tumor growth after splenectomy. This observation resulted from the lack of oncogene-induced senescence after splenectomy, which induces the formation of myeloid-derived suppressor cells (MDSC) in the spleen, bone marrow, and blood in animals with spleens [40]. Also, most of the studies performed in humans confirm that splenectomy patients have a higher risk of developing cancer [39].

On the other hand, tumor-associated macrophages and tumor-associated neutrophils infiltrate most human solid tumors and can promote cancer cell growth, metastasis, and angiogenesis through different mechanisms. They may originate partly from the spleen, and removal of the spleen before or after tumor development can effectively slow tumor growth [41]. Splenectomy seems to inhibit the early stages of tumorigenesis and reduces the rate of malignant transformation of benign tumors but does not prevent the progress of carcinogenesis, which was observed in female rats [40]. Stoth et al. also observed a decrease in hematogenous breast cancer lung metastases after splenectomy, which was attributed to changes in immune composition in the premetastatic niche in the lungs of breast cancer mice. They concluded that splenectomy affects not only the primary tumor but also changes in the immune microenvironment of the premetastases and metastases [42].

The spleen has been also discovered as involved in lipid metabolism. It acts as a reservoir for lipids; macrophages store fat through phagocytosis and the spleen exerts the LPL activity. Due to those reasons, splenectomy appeared to have a negative impact on lipid metabolism [11].

The current investigation is an attempt to extend our previous research efforts [27,43,44,45] to delve into the complexities of dietary supplements’ impact on cancerous processes, particularly focusing on the fatty acid (FA) profiles in the spleen. The preliminary studies revealed that the anti-carcinogenic properties of bitter melon and pomegranate supplements observed in vitro did not translate seamlessly into in vivo models, suggesting a more intricate interaction in living organisms. This complexity is further exemplified in our current study, where we explored the impact of CLnAs from pomegranate seed oil and bitter melon aqueous extract on the FA composition of the spleen and its potential implications in cancer biology.

As mentioned previously, no tumors of any kind developed in the DMBA-naive group, while in the “plus” group, the incidence of mammary carcinoma was 42%, 67%, and 100% in the CONplus, Mplus, Gplus, and GMplus groups, respectively. Furthermore, in the PSO-supplemented group, the first mammary tumors appeared much earlier than in the other groups, thus accelerating the tumorigenic process. In contrast to our previous findings, the treatment with DMBA, a known chemical carcinogen, did not result in noticeable changes in masses of most of the organs, irrespective of the dietary supplementation applied. Mean masses of livers and hearts in DMBA-naive groups were significantly higher compared to non-DMBA-treated animals [27]. On the other hand, we noticed significant variation in spleen mass within the groups of DMBA-treated animals (0.49 ± 0.07 g in CONplus, 0.63 ± 0.38 g in Mplus, 0.84 ± 0.60 g in Gplus and 0.70 ± 0.45 g in GMplus, respectively). This observation aligns with earlier studies, underscoring the significant impact of carcinogenic processes on internal organ physiology [46]. However, our current focus on the spleen’s FA profile revealed novel insights. We noted that while dietary supplements influenced the FA composition, these changes did not straightforwardly correlate with a reduction in the intensity of cancerous processes, as one might expect from the anti-carcinogenic reputation of these supplements in in vitro settings [27].

In our current study, we extend our investigation into the interactions of dietary supplements and cancerous processes, on the lipidomic profile of the organism, this time focusing on the FA profile in the spleen due to the involvement of the spleen in lipid metabolism. Similar to our previous findings in hepatic tissue, the spleen’s FA composition exhibited subtle modifications due to dietary supplementation, with a more pronounced impact evident in the context of the ongoing cancerous process [45].

Proper spleen functioning in the immune response of organisms depends on the adequate availability of PUFAs for incorporation into cell membranes. Dietary intake of PUFAs as well as endogenous synthesis in the spleen are both essential for proper PUFA availability. Dietary PUFAs are incorporated into lymphocytes and other leukocyte membranes and can modify the T-cell response to mitogens and antigens, whereas PUFA synthesis is directly involved in the regulation of lymphocyte activation and proliferation [47]. SFAs and PUFAs influence the effector and regulatory functions of innate and adaptive immune cells by changing membrane composition and fluidity as well as by acting through specific receptors. Also, impaired balance of ΣSFA/ΣPUFA, as well as Σn-6PUFA/Σn-3PUFA has significant consequences on immune system homeostasis [48]. PUFAs inhibit several lymphocyte functions including proliferation, interleukin-2 production, natural killer (NK) cell activity, and antigen presentation, which results in modification of immune response [49]. PUFAs play critical roles in T lymphocyte function and differentiation by maintaining the homeoviscosity of cell membranes and acting as substrates for the synthesis of lipid second messengers including oxylipins such as eicosanoids, hydroxyoctadecadienoic (HODEs) acid, hydroxyoctadecatrienoic acids (HOTrEs), and specialized pro-resolving mediators [50]. Dietary FAs affect T cells by altering their fate and modulating their effector functions. FAs passively diffuse across the T cell membrane and bind to cytoplasmic proteins called fatty acid binding proteins (FABPs). FABPs promote the nuclear localization of FAs and activate the nuclear receptor PPAR and promote its activation. In particular, T cells express FABP4 and 5, which have been reported to be involved in the uptake, transport, storage, and metabolism of FAs. In addition to this mechanism, FAs can bind to GPRs expressed on cell membranes; both T and B lymphocytes express GPR84, allowing the binding of medium-chain FAs such as lauric acid. Unsaturated FAs exert immunosuppressive effects on T cells. They can act on the early signaling of T cells and reduce T cell proliferation in a dose-dependent manner [48].

The application of bitter melon extract (BME) supplementation in healthy animals led to an increased content of specific FAs, much like the limited modifications observed in hepatic tissue. However, the introduction of pomegranate seed oil (PSO) resulted in more noticeable changes in the FA content compared to control groups without PSO supplementation. This pattern echoes our previous observations in serum and hepatic tissue, where dietary modifications alone did not result in significant differences in FA groups. However, the presence of a carcinogenic factor, such as DMBA, induced more substantial alterations in FA profiles, especially with the reduction of individual PUFA levels compared to healthy animals receiving supplementation [27,45].

Pomegranate and bitter melon may exert an immunomodulatory impact on organisms also by affecting spleen activity. Numerous studies confirmed the influence of those botanicals on the immunological system and inhibition of cytokines production, mainly pro-inflammatory ones, which results in the suppression of chronic inflammation involved in the pathogenesis of many diseases, including cancer [51,52,53,54,55,56,57]. However, most of them bind these properties with polyphenols present in mentioned botanicals not with CFA content.

We utilized Ag^+^-HPLC-DAD techniques with photodiode detection for the analysis of CFA isomers in the spleen. This analysis revealed that the highest concentration of CFA isomers was observed in healthy animals from the CON group. Conversely, the addition of BME, either alone or in conjunction with PSO, appeared to reduce CFA content, both CT and CD isomers, in the spleen tissue of healthy animals. This could be due to BME’s polyphenolic compounds altering the intestinal microbiota and subsequently affecting metabolic processes leading to CFA formation, a hypothesis supported by studies in ruminants and in rodents, where polyphenols modified the biohydrogenation of FA precursors to various CD isomers [43,58,59,60].

Interestingly, exposure to DMBA also influenced the concentration of CFA isomers in the spleen of supplemented animals, indicating that CLnA metabolism is altered under neoplastic conditions. Interestingly, in terms of the carcinogenic process, supplementation with PSO resulted in a significant elevation of levels of individual CT isomers compared to healthy counterparts. On the other hand, we did not observe significant differences within the group of rats with chemically induced cancer (except ttt and cct isomers levels), regardless of supplementation with PSO and BME, which aligns with alterations noted in serum PUFA profiles and our observations in the livers of rats with chemically induced cancer [27,45]. These findings reveal a complex interaction between dietary supplements and the spleen’s FA composition, especially under the influence of a carcinogenic agent like DMBA. Contrary to our initial hypothesis and previous observations in cardiac tissue [44], the FA profile in the spleen does not exhibit a straightforward protective or anti-carcinogenic pattern when exposed to CLnAs [41]. Instead, we observed nuanced alterations that suggest a more intricate role of these dietary components in the context of cancer.

Apart from significant insights into the FA profile in the spleen in the context of a cancerous environment, this study extends our previous research, where we explored the effects of CLA isomers and PSO on peroxidation indices (PI) and desaturase activities in different tissues [61,62]. The analysis of desaturase activities, enzymes critical in FA transformation, revealed some intriguing patterns. Desaturases, being oxidoreductases associated with endoplasmic reticulum membranes, play a pivotal role in introducing double bonds during FA biosynthesis. The effects of applied dietary supplements on desaturase activities, especially in the context of a neoplastic process, are relatively underexplored. Notably, in the BME-supplemented groups, we observed a decrease in delta-4 desaturase (D4D) and delta-5 desaturase activity (D5D), which might have resulted from competition between CLnAs (or its metabolites—CLA) and linoleic acid (LA) for the active sites of these enzymes or from the lowering effect of CFAs on arachidonic acid (AA) levels [63]. Intriguingly, we also noted a significant increase in D4D and D5D activity as an effect of supplementation with BME and PSO and exposure to DMBA. This finding is surprising considering that high dietary PUFA intake typically inhibits the expression of genes encoding desaturases [64]. The elevated desaturase indices in PSO-supplemented groups appear to be more a result of the coexisting neoplastic process than the dietary supplement per se. The findings from Kim et al. support this hypothesis, demonstrating a correlation between increased delta-9 desaturase (D9D) activity and the incidence of prostate cancer [65]. Our study reveals that dietary changes in healthy animals did not significantly impact the activities of the desaturases under investigation (Table 4). These patterns highlight the profound effect of the ongoing cancerous process on FA profiles and alterations in desaturase activities within the spleen. Changes in desaturase activity and FA profile in the spleen may also influence the cellular and humoral responses of the immunological system, which was confirmed in numerous studies [65,66,67].

Results of our study underline the significant role of the spleen in lipid metabolism, as an impact of dietary supplementation on spleen FA composition was observed. The crucial role in systemic lipid regulation of the spleen was also reported by Alberti et al., who found that splenectomy is associated with changes in lipid metabolism by an elevation of total cholesterol and low-density lipoprotein (LDL) levels while decreasing high-density lipoprotein (HDL) levels [68]. The alteration of LDL cholesterol due to the splenectomy was also observed by other authors [11,12], who linked this operation with the onset of atherosclerosis and other cardiovascular diseases. Wysocki et al. observed that in splenectomized patients, triglyceride (TG) levels are significantly higher, while in cases of splenomegaly, total cholesterol (TC) and LDL are notably decreased [69]. This highlights the importance of the spleen in systemic health and raises questions about how changes in spleen fatty acid content, as seen in our study, might influence overall metabolic health and disease risk. It has been confirmed that changes in desaturase activity connected with modification in FA profile influences also the cellular and humoral responses of the immunological system influencing the progression of many diseases including cancer [66,67,70,71]. It was also found that autogenous spleen tissue implants can revert the changes in lipid metabolism induced by splenectomy. Moreover, conservative spleen surgeries maintain normal lipid levels, suggesting the importance of spleen tissue preservation [68]. These findings are particularly interesting as they suggest that the presence of spleen tissue, even in a modified or rudimentary form, can maintain normal/physiological lipid metabolism. Moreover, it also indicates that maintaining spleen integrity (either physically or functionally through dietary means) is crucial for proper lipid metabolism, which could have implications for cancer progression and treatment. Tarantino et al. emphasizes the complexity of mechanisms involved in spleen function in lipid metabolism, highlighting the intricate nature of spleen function, also in the systemic immune response by immune modulation and peripheral tolerance. The spleen aids in clearing circulating apoptotic cells and producing antibodies in the white pulp [72]. These observations seem to be essential also in the context of cancer, in which immunology is essentially altered. It may lead to the assumption that maintaining the good condition of the spleen and its lipid profile modification may affect the role of this internal organ in anticancerogenic processes.

The chemometric analysis we applied to our data solidified observations indicating that the presence of a pathological process is the main trigger of changes in FA and CFA profiles in the spleen than the administration of PSO. Notably, the alterations in lipid profiles induced by applied supplementation showed considerable variation between physiological and pathological states.

## 5. Conclusions

The presented investigation of the impact of dietary supplements on fatty acid profiles of the spleen, particularly under cancerous conditions, reveals complex and nuanced interactions. We observed that supplements like bitter melon extract and pomegranate seed oil lead to significant changes in the spleen’s fatty acid composition. While BME supplementation led to increased specific FAs in healthy animals, supplementation with PSO resulted in more pronounced alterations. These changes were most significant in the presence of DMBA, suggesting that the spleen’s FA composition is significantly modified throughout the cancerous process. This study also highlights the spleen’s role in systemic lipid metabolism (together with the liver) and its potential implications in cancer progression and treatment. The altered conjugated fatty acid profiles and desaturase activities in the spleen, especially under neoplastic conditions, emphasize the intricate relationship between diet, disease, and metabolic regulation. These findings provide essential insights into how dietary modifications can influence lipid metabolism in the spleen, underscoring the importance of considering these factors in cancer research and treatment strategies.

## Figures and Tables

**Figure 1 cancers-16-00479-f001:**
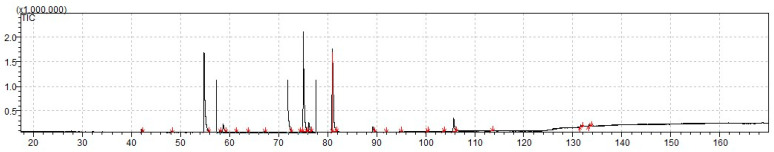
Exemplary GC-MS chromatogram of spleen sample.

**Figure 2 cancers-16-00479-f002:**
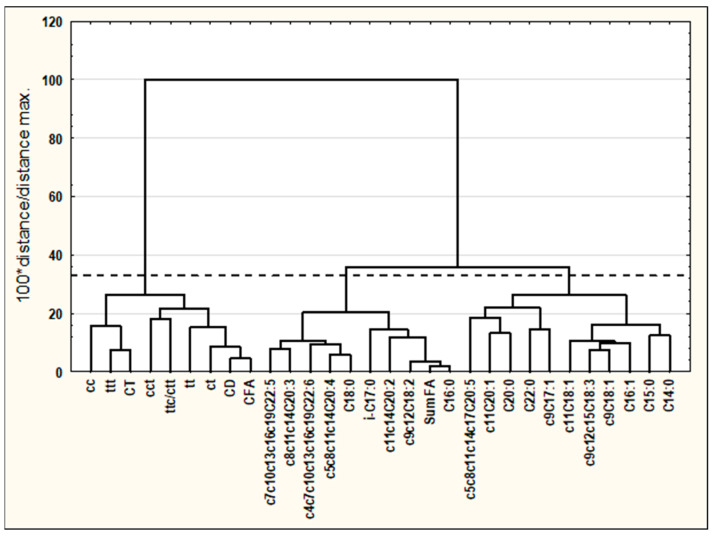
Dendrogram of similarities of examined features (variables).

**Figure 3 cancers-16-00479-f003:**
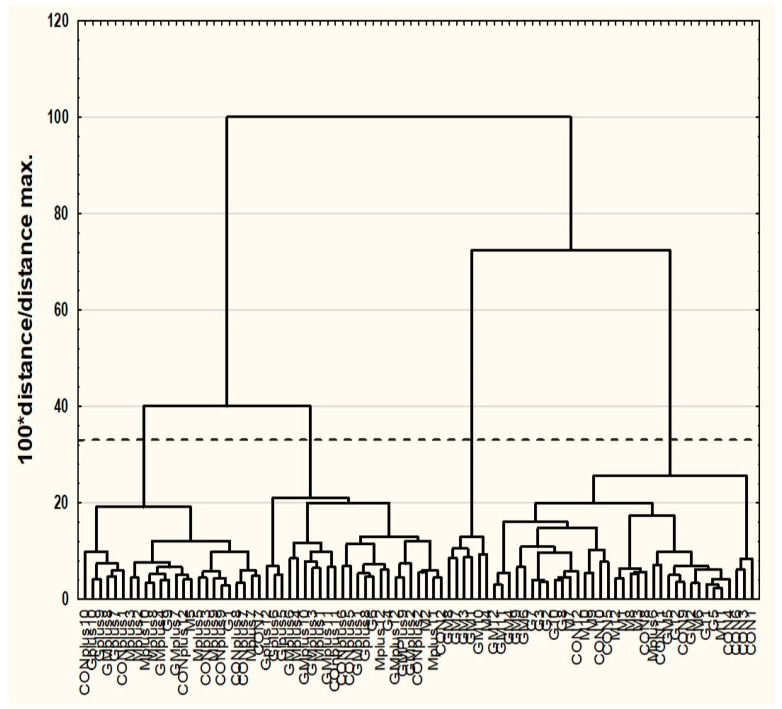
Dendrogram of similarities of examined objects (individuals).

**Figure 4 cancers-16-00479-f004:**
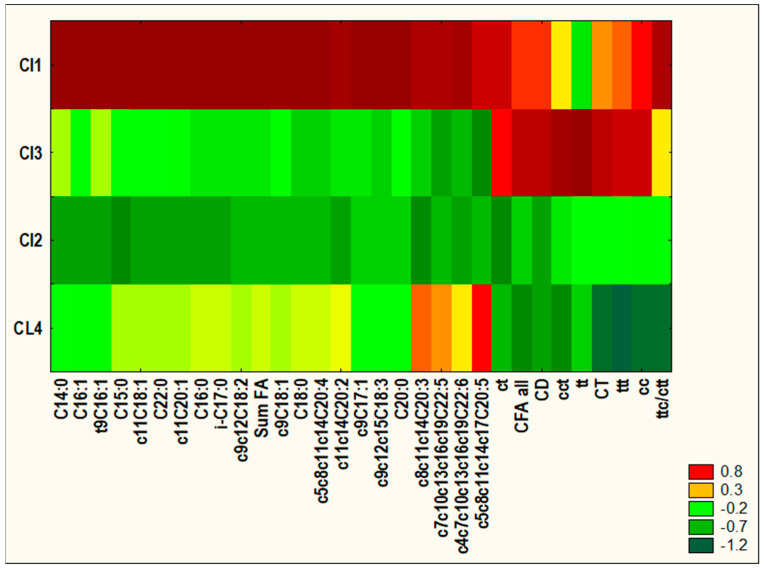
Similarity analysis performed by grouping of features and objects to prepare heat map.

**Table 1 cancers-16-00479-t001:** Individual fatty acids content (µg/g fresh tissue) and the content sums (mg/g fresh tissue) of saturated (SFA), monounsaturated (MUFA), polyunsaturated (PUFA) fatty acids, and all assayed fatty acids (Total FA) in spleen of rats fed the control (CON) and experimental diets.

Fatty Acids (µg/g)	CON	M	G	GM	CONplus	Mplus	Gplus	GMplus	*p* Value
C14:0	182.0 ± 69.1	228.8 ± 72.5 ^a,b^	130.0 ± 47.3 ^a^	259.2 ± 145.9 ^c^	172.9 ± 100.4	123.1 ± 37.2 ^b,c^	159.5 ± 71.4	211.7 ± 125.9	0.0020
C15:0	96.5 ± 44.5	109.0 ± 25.2 ^a^	82.8 ± 28.4	114.5 ± 46.8	81.0 ± 27.4	64.0 ± 20.0 ^a^	73.6 ± 35.9	99.9 ± 21.2	0.0085
C16:0	5839 ± 1302 ^a^	6129 ± 1617 ^b,c^	5374 ± 982 ^d^	8537 ± 2614 ^e,f,g,h^	3445 ± 713 ^a,b,d,e^	4021 ± 951 ^c,f^	4219 ± 1919 ^g^	4469 ± 856 ^h^	<0.0001
i-C17:0	128.5 ± 56.2 ^a^	103.7 ± 37.0	99.5 ± 25.8	134.3 ± 53.1 ^b^	66.1 ± 31.3 ^a,b^	79.3 ± 33.3	76.7 ± 29.6	96.7 ± 28.9	0.0018
C18:0	3830 ± 1118	4572 ± 1416 ^a,b,c,d^	3970 ± 507 ^e^	5919 ± 1377 ^f,g,h,i^	2683 ± 526 ^a,f^	2666 ± 393 ^b,e,g^	2614 ± 757 ^c,h^	2872 ± 519 ^d,i^	<0.0001
C20:0	30.6 ± 13.5	48.8 ± 19.6 ^a^	29.3 ± 10.9	49.5 ± 26.8	44.6 ± 15.1 ^b^	26.2 ± 11.8	20.2 ± 11.5 ^a,b^	31.8 ± 18.6	0.0013
C22:0	35.9 ± 12.5	34.8 ± 15.9	27.2 ± 13.3	55.0 ± 26.7 ^a,b^	28.7 ± 25.4	23.4 ± 16.9 ^a^	21.5 ± 5.5 ^b^	29.7 ± 15.6	0.0041
SFA (mg/g)	10.14 ± 2.0 ^a,b^	11.21 ± 3.00 ^c,d^	9.71 ± 1.57 ^e^	15.07 ± 4.20 ^f,g,h,i^	6.52 ± 1.32 ^a,c,e,f^	7.00 ± 1.20 ^b,d,g^	7.18 ± 2.73 ^h^	7.81 ± 1.40 ^i^	<0.0001
ΣC16:1^#^	503.7 ± 322.2	405.0 ± 215.4	321.6 ± 175.1 ^a^	694.2 ± 480.3 ^a,b,c^	248.1 ± 136.0 ^b^	302.1 ± 133.6 ^c^	437.0 ± 318.0	434.4 ± 276.0	0.0083
c9C17:1	71.2 ± 34.7 ^a^	53.9 ± 21.9 ^b^	32.3 ± 8.5 ^a,b,c^	87.4 ± 37.2 ^c,d^	47.3 ± 8.9	47.0 ± 10.6	41.9 ± 2.5	40.2 ± 11.5 ^d^	<0.0001
c6C18:1	28.1 ± 15.7	25.1 ± 14.3	n.d.	27.7 ± 9.0	35.6 ± 25.1	23.7 ± 12.9	19.4 ± 9.7	21.8 ± 10.6	0.6736
c7C18:1	23.9 ± 10.7	28.1 ± 11.6	n.d.	38.8 ± 22.3 ^a^	20.4 ± 15.5	14.4 ± 8.1 ^a^	13.9 ± 7.8	15.8 ± 8.9	0.0025
c9C18:1	3702 ± 2152 ^a^	2923 ± 1265 ^b^	2171 ± 686	4254 ± 2214 ^b,d^	1480 ± 564 ^a,b,c^	1968 ± 565 ^d^	1750 ± 865	1988 ± 528	<0.0001
c11C18:1	634.4 ± 197.0	649.9 ± 227.7	488.1 ± 86.5	718.8 ± 232.9 ^a^	459.5 ± 95.4 ^a^	522.6 ± 144.4	486.5 ± 245.0	489.3 ± 83.2	0.0098
c11C20:1	77.1 ± 20.9	95.0 ± 34.3	69.6 ± 15.1	91.0 ± 39.1	69.2 ± 23.5	63.8 ± 18.4	58.8 ± 37.7	71.3 ± 24.0	<0.0001
MUFA (mg/g)	5.0 ± 2.65 ^a^	4.18 ± 1.67	3.08 ± 0.93	5.91 ± 2.96 ^b,c^	2.36 ± 0.76 ^a,b^	2.94 ± 0.79 ^c^	2.81 ± 1.36	3.06 ± 0.77	0.0002
c9c12C18:2	7441 ± 2227 ^a,b^	7443 ± 2033 ^c,d^	6628 ± 1167 ^e^	11,752 ± 4059 ^f,g,h,i^	3536 ± 1195 ^a,c,e,f^	4633 ± 875 ^b,d,g^	5335 ± 2766 ^h^	5068 ± 1194 ^i^	<0.0001
c9c12c15C18:3	244.2 ± 202.2 ^a,b,c^	145.5 ± 64.6	103.0 ± 44.8	292.6 ± 193.2 ^d,e,f^	126.8 ± 41.0	71.4 ± 50.7 ^a,d^	62.3 ± 35.7 ^b,e^	68.4 ± 35.0 ^c,f^	<0.0001
c9t11C18:2	n.d.	n.d.	67.2 ± 29.4	104.5 ± 53.1 ^a,b,c^	38.2 ± 25.2 ^a^	45.1 ± 29.1 ^b^	30.2 ± 18.1 ^c^	48.1 ± 21.6	0.0004
c11c14C20:2	80.2 ± 25.6 ^a^	78.8 ± 32.1 ^b^	85.6 ± 28.2 ^c^	113.4 ± 35.4 ^d,e,f^	38.1 ± 12.4 ^a,b,c,d^	52.4 ± 18.6 ^e^	65.2 ± 24.5 ^f^	61.3 ± 15.9	<0.0001
c8c11c14C20:3	117.5 ± 31.2 ^a,b,c,d^	171.8 ± 43.5 ^a,e,f,g,h^	158.7 ± 25.8 ^b,I,j,k,l^	191.3 ± 21.1 ^c,m,n,o,p^	86.8 ± 27.4 ^e,i,m^	75.5 ± 21.8 ^d,f,j,n^	88.7 ± 34.6 ^g,k,o^	88.4 ± 36.8 ^h,l,p^	<0.0001 *
c5c8c11c14C20:4	3384 ± 1083 ^a^	4401 ± 1205 ^b,c^	4256 ± 555 ^d,e^	5665 ± 925 ^a,f,g,h,i^	3394 ± 490 ^f^	3069 ± 574 ^b,d,g^	2580 ± 655 ^c,e,h^	3327 ± 393 ^i^	<0.0001
c5c8c11c14c17C20:5	89.3 ± 56.0	193.2 ± 122.6 ^a,b,c^	96.7 ± 30.1 ^d^	107.3 ± 52.9 ^e^	103.9 ± 49.1 ^f^	59.4 ± 35.2 ^a^	27.7 ± 14.4 ^b,d,e,f^	46.8 ± 32.2 ^c^	<0.0001
c7c10c13c16c19C22:5	403.0 ± 122.6	609.8 ± 157.1 ^a,b,c,d^	516.0 ± 63.7 ^e,f,g^	742.9 ± 159.4 ^h,I,j,k^	227.3 ± 28.0 ^a,h^	210.8 ± 58.9 ^b,e,i^	164.2 ± 39.0 ^c,f,j^	198.7 ± 55.6 ^d,g,k^	<0.0001
c4c7c10c13c16c19C22:6	316.3 ± 79.2	362.9 ± 94.5	358.9 ± 74.7	504.1 ± 92.8	247.9 ± 67.8	265.5 ± 74.0	194.4 ± 71.2	244.8 ± 71.1	<0.0001 *
PUFA (mg/g)	12.01 ± 2.94 ^a^	13.37 ± 3.28 ^b,c^	12.29 ± 1.59 ^d,e^	19.47 ± 5.08 ^f,g,h,i^	7.84 ± 1.48 ^a,b,d,f^	8.50 ± 1.36 ^c,e,g^	8.55 ± 3.07 ^h^	9.14 ± 1.39 ^i^	<0.0001
Total FA (mg/g)	27.52 ± 6.06 ^a,b^	28.90 ± 7.49 ^c,d^	25.07 ± 4.16 ^e^	40.43 ± 12.38 ^f,g,h,i^	16.45 ± 3.68 ^a,c,e,f^	18.89 ± 3.91 ^b,d,g^	19.39 ± 8.67 ^h^	19.63 ± 3.51 ^i^	<0.0001

Data of FA content in fresh tissue are shown as mean values ± standard deviation (SD). *p* value ≤ 0.05—significant differences among groups in one-way ANOVA (*) or Kruskal–Wallis test. Values sharing a letter in one row are significantly different (*p* < 0.05) in RIR Tukey test (*) or multiple comparison test. n.d.—not detected (i.e., below of the limit of detection (LOD)) ΣC16:1^#^—sum of two C16:1 isomers detected in spleen samples was presented, due to inability of their exact identification.

**Table 2 cancers-16-00479-t002:** Fatty acids share (% of total fatty acids) in spleens of control (CON) and experimental groups.

Fatty Acids (%)	CON	M	G	GM	CONplus	Mplus	Gplus	GMplus	*p* Value
C14:0	0.65 ± 0.17	0.90 ± 0.37 ^a^	0.51 ± 0.12 ^a,b,c^	0.61 ± 0.18	0.89 ± 0.30 ^b^	0.66 ± 0.19	0.84 ± 0.27	0.85 ± 0.17 ^c^	0.0001
C15:0	0.37 ± 0.09 ^a^	0.39 ± 0.09 ^b^	0.33 ± 0.10 ^c,d^	0.28 ± 0.06 ^e,f^	0.48 ± 0.09 ^g,c,e^	0.34 ± 0.09 ^h,g^	0.38 ± 0.13	0.51 ± 0.07 ^a,b,c,d,f,h^	<0.0001 *
C16:0	21.25 ± 1.62	21.18 ± 0.98	21.38 ± 0.50	21.12 ± 0.56	20.50 ± 0.64 ^a^	21.20 ± 1.48	21.99 ± 0.50	22.86 ± 1.79 ^a^	0.0024
i-C17:0	0.42 ± 0.10	0.37 ± 0.05	0.40 ± 0.03	0.33 ± 0.05 ^a,b^	0.39 ± 0.14	0.45 ± 0.12 ^a^	0.42 ± 0.13	0.46 ± 0.09 ^b^	0.0090
C18:0	14.02 ± 3.16	15.82 ± 2.01	15.92 ± 0.62	14.97 ± 1.42	15.98 ± 1.97	15.11 ± 2.00	14.43 ± 2.93	14.29 ± 1.06	0.0736
C20:0	0.13 ± 0.06	0.17 ± 0.06	0.11 ± 0.03 ^a^	0.12 ± 0.05	0.25 ± 0.09 ^a,b^	0.14 ± 0.07	0.12 ± 0.07 ^b^	0.15 ± 0.08	0.0065
C22:0	0.12 ± 0.02	0.12 ± 0.05	0.11 ± 0.05	0.14 ± 0.06	0.11 ± 0.05	0.11 ± 0.05	0.13 ± 0.06	0.14 ± 0.07	0.7955
SFA	36.97 ± 1.82 ^a^	38.94 ± 1.46	38.76 ± 0.52	37.56 ± 1.41	38.59 ± 2.32	38.02 ± 1.98	38.31 ± 3.32	39.27 ± 1.16 ^a^	0.0345
ΣC16:1 ^#^	1.83 ± 1.12	1.25 ± 0.34	1.24 ± 0.46	1.56 ± 0.67	1.37 ± 0.54	1.57 ± 0.52	2.00 ± 1.05	2.00 ± 0.81	0.0686
c9C17:1	0.25 ± 0.10 ^a^	0.16 ± 0.05 ^b^	0.10 ± 0.02 ^a,c,d,e^	0.18 ± 0.06	0.27 ± 0.05 ^b,c^	0.24 ± 0.09 ^d^	0.16 ± 0.02	0.21 ± 0.07 ^e^	<0.0001
c6C18:1	0.09 ± 0.04	0.09 ± 0.06	n.d.	0.07 ± 0.02 ^a^	0.18 ± 0.10 ^a^	0.11 ± 0.05	0.12 ± 0.09	0.10 ± 0.06	0.0240
c7C18:1	0.09 ± 0.03	0.10 ± 0.05	n.d.	0.09 ± 0.04	0.10 ± 0.07	0.08 ± 0.05	0.08 ± 0.05	0.08 ± 0.05	0.8734
c9C18:1	13.21 ± 6.85	9.07 ± 1.23	8.53 ± 1.39	9.96 ± 2.32	9.82 ± 4.60	10.47 ± 2.21	9.72 ± 2.11	10.74 ± 2.96	0.1987
c11C18:1	2.31 ± 0.50	2.22 ± 0.35	1.95 ± 0.13 ^a,b,c^	1.77 ± 0.15 ^d,e,f,g^	2.76 ± 0.16 ^a,d^	2.75 ± 0.31 ^b,e^	2.46 ± 0.24 ^f^	2.60 ± 0.52 ^c,g^	<0.0001
c11C20:1	0.28 ± 0.07 ^a^	0.33 ± 0.09	0.28 ± 0.05 ^b^	0.23 ± 0.08 ^c,d^	0.41 ± 0.10 ^a,b,c^	0.35 ± 0.11 ^d^	0.29 ± 0.12	0.34 ± 0.10	0.0003 *
MUFA	18.06 ± 8.32	13.22 ± 1.66	12.10 ± 1.78 ^a^	13.87 ± 3.02	14.92 ± 4.96	15.57 ± 2.72 ^a^	14.83 ± 3.20	16.06 ± 3.95	0.0389
c9c12C18:2	26.72 ± 3.09 ^a^	25.70 ± 1.41 ^b^	26.85 ± 0.81 ^c^	28.81 ± 1.71 ^b,d,e^	22.83 ± 1.73 ^a,c,d,f^	24.68 ± 2.16 ^e^	27.01 ± 2.54 ^f^	26.31 ± 1.25	<0.0001
c9c12c15C18:3	0.87 ± 0.68 ^a,b^	0.49 ± 0.13	0.40 ± 0.12	0.66 ± 0.27 ^c^	0.76 ± 0.27 ^d,e,f^	0.38 ± 0.29 ^d^	0.27 ± 0.12 ^a,e^	0.30 ± 0.13 ^b,c,f^	<0.0001
c9t11C18:2	n.d.	n.d.	0.26 ± 0.09	0.29 ± 0.14	0.26 ± 0.18	0.24 ± 0.16	0.17 ± 0.11	0.23 ± 0.10	0.3948
c11c14C20:2	0.30 ± 0.10	0.27 ± 0.06	0.34 ± 0.09 ^a^	0.27 ± 0.04	0.23 ± 0.06 ^a,b^	0.28 ± 0.09	0.35 ± 0.05 ^b^	0.32 ± 0.09	0.0129
c8c11c14C20:3	0.44 ± 0.14 ^a^	0.61 ± 0.17 ^b,c^	0.64 ± 0.10 ^a,d,e^	0.49 ± 0.12	0.52 ± 0.14	0.40 ± 0.09 ^b,d^	0.50 ± 0.20	0.42 ± 0.17 ^c,e^	0.0003 *
c5c8c11c14C20:4	12.63 ± 4.21 ^a^	16.00 ± 1.43	17.10 ± 1.49	14.61 ± 2.37 ^b^	18.44 ± 4.02 ^a,b^	17.34 ± 2.39	14.72 ± 3.96	15.59 ± 2.66	0.0009
c5c8c11c14c17C20:5	0.28 ± 0.15 ^a^	0.73 ± 0.53 ^b,c^	0.39 ± 0.11 ^d,e^	0.27 ± 0.10 ^f^	0.63 ± 0.29 ^a,f,g,h^	0.29 ± 0.14	0.13 ± 0.05 ^b,d,g^	0.18 ± 0.09 ^c,e,h^	<0.0001
c7c10c13c16c19C22:5	1.67 ± 0.73	2.14 ± 0.43 ^a,b,c,d^	2.08 ± 0.21 ^e,f,g,h^	1.91 ± 0.41 ^h,I,j^	1.35 ± 0.24 ^a,^	1.12 ± 0.25 ^b,e,h^	1.00 ± 0.45 ^c,f,i^	0.99 ± 0.31 ^d,g,j^	<0.0001
c4c7c10c13c16c19C22:6	1.25 ± 0.37	1.27 ± 0.23	1.45 ± 0.29	1.31 ± 0.29	1.48 ± 0.36	1.40 ± 0.26	1.06 ± 0.30	1.20 ± 0.37	0.0874 *
PUFA	44.16 ± 6.40	47.22 ± 2.18	49.51 ± 1.57 ^a,b,c^	48.62 ± 1.88	46.48 ± 4.11	46.13 ± 2.24 ^a^	45.21 ± 3.25 ^b^	45.53 ± 3.29 ^c^	0.0034

Data are shown as mean values ± standard deviation (SD). *p* value ≤ 0.05—significant differences among groups in one-way ANOVA (*) or Kruskal–Wallis test. Values sharing a letter in one row are significantly different (*p* < 0.05) in RIR Tukey test (*) or multiple comparison test. n.d.—not detected (<LOD); i-C17:0—iso-C17, ΣC16:1 ^#^—sum of two C16:1 isomers detected in spleen samples was presented, due to inability of their exact identification.

**Table 3 cancers-16-00479-t003:** Content (ug/g) of assayed fatty acids and values of indices calculated based on fatty acid contents in rats’ spleen.

	CON	M	G	GM	CONplus	Mplus	Gplus	GMplus	*p* Value
ΣSFA	10,143 ± 2094 ^a,b^	11,226 ± 2998 ^c,d^	9713 ± 1573 ^e^	15,069 ± 4202 ^f,g,h,i^	6521 ± 1320 ^a,c,e,f^	7003 ± 1197 ^b,d,g^	7184 ± 2731 ^h^	7811 ± 1403 ^i^	<0.0001
ΣMUFA	5040 ± 2647 ^a^	4180 ± 1669	3082 ± 929	5912 ± 2960 ^b,c^	2360 ± 759 ^a,b^	2941 ± 793 ^c^	2807 ± 1365	3060 ± 770	0.0002
ΣPUFA	12,007 ± 2942 ^a^	13,367 ± 3280 ^b,c^	12,291 ± 1587 ^d,e^	19,475 ± 5081 ^f,g,h,i^	7835 ± 1482 ^a,b,d,f^	8497 ± 1356 ^c,e,g^	8545 ± 3069 ^h^	9144 ± 1391 ^i^	<0.0001
Σn-3 PUFA	1053 ± 145 ^a,b,c^	1311 ± 313 ^d,e,f,g^	1075 ± 143 ^h,i,j^	1647 ± 400 ^k,l,m,n^	706 ± 146 ^d,k^	607 ± 157 ^a,e,h,l^	448 ± 94 ^b,f,i,m^	559 ± 113 ^c,g,j,n^	<0.0001
Σn-6 PUFA	11,023 ± 2940 ^a^	12,095 ± 3156 ^b,c^	11,128 ± 1615 ^d^	17,722 ± 4909 ^e,f,g,h^	7055 ± 1560 ^a,b,d,e^	7830 ± 1293 ^c,f^	8069 ± 3101 ^g^	8545 ± 1425 ^h^	<0.0001
Σn-3 PUFA/Σn-6 PUFA	0.101 ± 0.029 ^a^	0.110 ± 0.021 ^b,c,d^	0.097 ± 0.007 ^e,f^	0.094 ± 0.008	0.102 ± 0.018 ^g,h^	0.078 ± 0.018 ^b^	0.061 ± 0.020 ^c,e,g^	0.066 ± 0.016 ^a,d,f,h^	<0.0001
ΣLPUFA	4390 ± 1296 ^a^	5818 ± 1433 ^b,c^	5472 ± 683 ^d,e^	7324 ± 1188 ^f,g,h,i,a^	4098 ± 580 ^f^	3733 ± 686 ^b,d,g^	3120 ± 763 ^c,e,h^	3967 ± 446 ^i^	<0.0001
Σn-6 LPUFA	3582 ± 1124 ^a^	4652 ± 1258 ^b,c^	4500 ± 591 ^d,e^	5970 ± 969 ^f,g,h,i,a^	3519 ± 507 ^f^	3197 ± 593 ^b,d,g^	2734 ± 694 ^c,e,h^	3477 ± 419 ^i^	<0.0001
Σn-3 LPUFA	809 ± 229	1166 ± 275 ^a,b,c,d^	972 ± 116 ^e,f,g^	1354 ± 251 ^h,i,j,k^	579 ± 114 ^a,h^	536 ± 134 ^b,e,i^	386 ± 81 ^c,f,j^	490 ± 103 ^d,g,k^	<0.0001
A-SFA	6021 ± 1362 ^a^	6357 ± 1642 ^b,c^	5504 ± 1022 ^d^	8796 ± 2748 ^e,f,g,h^	3618 ± 758 ^a,b,d,e^	4144 ± 964 ^c,f^	4378 ± 1977 ^g^	4681 ± 949 ^h^	<0.0001
A-SFA/ΣFA	0.219 ± 0.017	0.220 ± 0.009	0.219 ± 0.005	0.217 ± 0.006	0.221 ± 0.023	0.219 ± 0.015	0.224 ± 0.013	0.238 ± 0.020	0.0621
T-SFA	9851 ± 2030 ^a,b^	10,930 ± 2941 ^c,d^	9474 ± 1523 ^e^	14,715 ± 4076 ^f,g,h,i^	6301 ± 1262 ^a,c,e,f^	6810 ± 1160 ^b,d,g^	6992 ± 2675 ^h^	7553 ± 1349 ^i^	<0.0001
T-SFA/ΣFA	0.359 ± 0.018 ^a^	0.378 ± 0.015	0.378 ± 0.005	0.367 ± 0.014	0.386 ± 0.031 ^a^	0.363 ± 0.020	0.369 ± 0.027	0.385 ± 0.021	0.0162
_index_A^SFA^	0.384 ± 0.024	0.401 ± 0.020	0.384 ± 0.015	0.378 ± 0.015 ^a^	0.408 ± 0.045	0.396 ± 0.046	0.422 ± 0.039	0.434 ± 0.051 ^a^	0.0041
_index_T^SFA^	9813 ± 3109	11,011 ± 3499 ^a^	10,208 ± 1561 ^b^	15,582 ± 4013 ^c,d,e,f^	6516 ± 1531 ^a,b,c^	7422 ± 1336 ^d^	8351 ± 3590 ^e^	8637 ± 1709 ^f^	<0.0001
ΣSFA/ΣUFA	0.601 ± 0.058	0.640 ± 0.037	0.631 ± 0.017	0.599 ± 0.032	0.638 ± 0.048	0.614 ± 0.050	0.636 ± 0.050	0.638 ± 0.028	0.0589
ΣSFA/ΣPUFA	0.855 ± 0.088 ^a^	0.839 ± 0.053	0.788 ± 0.040	0.771 ± 0.033 ^a,b^	0.834 ± 0.089	0.826 ± 0.069	0.835 ± 0.065	0.853 ± 0.068 ^b^	0.0035
ΣSFA/ΣMUFA	2.422 ± 0.927	2.882 ± 0.644	3.258 ± 0.464 ^a^	2.817 ± 0.618	2.906 ± 0.589	2.463 ± 0.467 ^a^	2.792 ± 0.658	2.619 ± 0.409	0.0362
ΣSFA/ΣFA	0.370 ± 0.019 ^a,b^	0.388 ± 0.015	0.388 ± 0.006	0.376 ± 0.014	0.399 ± 0.031 ^b^	0.373 ± 0.020	0.379 ± 0.029	0.398 ± 0.021 ^a^	0.0076
ΣMUFA/ΣFA	0.181 ± 0.083	0.143 ± 0.043	0.121 ± 0.018	0.139 ± 0.030	0.147 ± 0.056	0.156 ± 0.027	0.143 ± 0.035	0.156 ± 0.032	0.0525
^C18:0^Δ9_index_	0.464 ± 0.169	0.382 ± 0.094	0.347 ± 0.043	0.397 ± 0.075	0.348 ± 0.079	0.420 ± 0.069	0.383 ± 0.085	0.406 ± 0.068	0.0552
^ΣΔ9,6,5,4^FA_index_	0.635 ± 0.023	0.620 ± 0.014	0.621 ± 0.006	0.633 ± 0.013	0.623 ± 0.017	0.629 ± 0.019	0.622 ± 0.019	0.623 ± 0.009	0.1638
^n6ElongC20/C18^ index	0.011 ± 0.004	0.010 ± 0.002	0.013 ± 0.004	0.010 ± 0.003	0.011 ± 0.003	0.011 ± 0.004	0.013 ± 0.003	0.012 ± 0.003	0.2302
^n3ElongC22/C20^index	0.828 ± 0.057	0.769 ± 0.128	0.844 ± 0.038	0.877 ± 0.045 ^a^	0.700 ± 0.096 ^a^	0.786 ± 0.094	0.851 ± 0.081	0.819 ± 0.090	0.0015
Δ4_index_	0.443 ± 0.039	0.373 ± 0.027 ^a,b,c,d^	0.408 ± 0.054 ^e,f,g^	0.407 ± 0.050 ^h,i^	0.513 ± 0.083 ^a,e^	0.556 ± 0.054 ^b,f,h^	0.529 ± 0.128 ^c^	0.548 ± 0.094 ^d,g,i^	<0.0001
Δ5_index_	0.965 ± 0.006	0.962 ± 0.007 ^a,b,c^	0.964 ± 0.005 ^d,e^	0.967 ± 0.004	0.975 ± 0.007 ^a,d^	0.976 ± 0.006 ^b,e^	0.967 ± 0.008	0.974 ± 0.010 ^c^	<0.0001
h/H-Ch	2.566 ± 0.171	2.511 ± 0.128	2.540 ± 0.089	2.617 ± 0.095 ^a^	2.498 ± 0.184	2.491 ± 0.279	2.370 ± 0.221	2.342 ± 0.198 ^a^	0.0090

Data are shown as mean values ± standard deviation (SD). *p* value ≤0.05—significant differences among groups in Kruskal–Wallis test. Values sharing a letter in one row are significantly different (*p* < 0.05) in multiple comparison test. A-SFA = C12:0 + C14:0 + C16:0; T-SFA = C14:0 + C16:0 + C18:0; _index_A^SFA^= (C12:0 + 4 × C14:0 + C16:0)/(ΣMUFA + Σn6 PUFA + Σn3 PUFA); _index_T^SFA^ = (C14:0 + C16:0 + C18:0)/[(0.5 × ΣMUFA + 0.5 × Σn-6PUFA + 3 × Σn-3PUFA)/Σn-6PUFA)]; ^C18:0^∆9_index_ = c9C18:1/(c9C18:1 + C18:0); ^Σ∆9,6,5,4^FA_index_ = (ΣMUFA + ΣPUFA)/(C16:0 + C18:0 + C20:0 + C22:0 + C24:0 +ΣMUFA + ΣPUFA); ^n6ElongC20/C18^index = c11c14C20:2/(c11c14C20:2 + LA); ^n3ElongC22/C20^index = DPA/(DPA + EPA); Δ4_index_ = DHA/(DHA + DPA); Δ5_index_ = AA/(AA + c8c11c14C20:3); h/H-Ch = (c7C18:1 + c9C18:1 + c12C18:1 + c14C18:1 + c11C20:1 + 13C22:1 + LA + ALA + c6c9c12C18:3 + AA + c11c14C20:2 + EPA + c7c10c13c16C22:4+ DPA)/(C14:0 + C16:0).

**Table 4 cancers-16-00479-t004:** Content of CFA isomers in rats’ spleen (µg/g fresh tissue).

µg/g Sample	CON	M	G	GM	CONplus	Mplus	Gplus	GMplus	*p* Value
CFAs	1008 ± 264 ^a,b,c^	491.1 ± 119.7 ^a^	599.6 ± 155.7 ^b^	690.7 ± 177.6	685.4 ± 163.1	646.0 ± 289.0 ^c^	826.9 ± 130.0	766.8 ± 287.1	<0.0001
CDs	805.3 ± 241.5 ^a^	483.2 ± 117.5 ^a,b^	585.5 ± 155.6	592.6 ± 156.5	602.6 ± 145.7	556.2 ± 217.4	729.5 ± 122.6 ^b^	605.7 ± 235.0	0.0040
tt	294.9 ± 149.2	206.3 ± 47.1 ^a^	207.1 ± 57.7 ^b^	205.1 ± 41.1	280.5 ± 121.7	224.2 ± 76.2	306.8 ± 54.5 ^a,b^	271.9 ± 94.2	0.0075
ct	478.4 ± 81.3 ^a,b,c,d^	258.7 ± 73.8 ^a^	354.4 ± 106.4	355.6 ± 140.6	292.7 ± 42.4 ^b^	290.3 ± 146.2 ^c^	400.3 ± 89.0	297.0 ± 166.0 ^d^	0.0004
cc	32.01 ± 26.07	18.28 ± 8.72 ^a^	24.65 ± 11.16	31.80 ± 13.23	29.45 ± 15.37	41.63 ± 20.21 ^a^	33.64 ± 12.57	36.86 ± 18.56	0.0213
CTs	187.2 ± 62.8 ^a,b,c^	7.89 ± 3.50 ^a,d,e,f,g^	31.52 ± 26.82 ^b,h^	98.18 ± 50.47 ^d^	61.5 ± 41.8 ^c^	89.86 ± 101.26 ^e^	97.32 ± 26.60 ^f^	161.1 ± 56.6 ^g,h^	<0.0001
ttt	168.0 ± 58.1 ^a,b,c,d^	3.55 ± 1.98 ^a,e,f,g^	27.52 ± 21.82 ^b,h^	81.77 ± 51.75 ^e^	39.5 ± 16.8 ^c,i^	55.36 ± 54.43 ^d^	67.94 ± 21.35 ^f^	145.4 ± 52.5 ^g,h,i^	<0.0001
ttc/ctt	11.41 ± 9.80 ^a^	1.54 ± 0.65 ^a,b,c,d^	4.24 ± 2.54	10.73 ± 11.37	9.53 ± 11.59 ^b^	6.33 ± 6.00	10.07 ± 5.24 ^c^	6.76 ± 5.47 ^d^	0.0004
cct	7.20 ± 5.38 ^a^	2.88 ± 1.93 ^b^	1.44 ± 0.53 ^a,c,d^	4.64 ± 2.60 ^d^	5.02 ± 5.49 ^e^	3.34 ± 1.41 ^f^	21.69 ± 16.23 ^b,d,e,f^	3.65 ± 1.56	<0.0001

Data are shown as mean values ± standard deviation (SD). *p* value ≤ 0.05—significant differences among groups in Kruskal–Wallis test. Values sharing a letter in one row are significantly different (*p* < 0.05) in multiple comparison test; CFAs—conjugated fatty acids; CDs—conjugated dienes; CTs—conjugated trienes.

**Table 5 cancers-16-00479-t005:** Percentage (%) profile of CFA isomers in spleen of rats fed the control and experimental groups.

	CON	M	G	GM	CONplus	Mplus	Gplus	GMplus	*p* Value
(% CFAs)									
CDs	79.5 ± 4.4 ^a,b^	98.4 ± 0.5 ^a,c,d,e,f,g^	97.6 ± 6.9 ^b^	85.9 ± 6.6 ^c^	88.5 ± 8.8 ^d^	87.5 ± 9.8 ^e^	88.1 ± 3.0 ^f^	78.8 ± 2.7 ^g^	<0.0001
tt	28.0 ± 5.4 ^a,b^	42.2 ± 4.4 ^a,c^	35.1 ± 7.0	30.7 ± 6.3 ^b,c^	40.1 ± 9.2	37.1 ± 9.4	37.2 ± 3.9	37.1 ± 8.7	<0.0001 *
ct	48.4 ± 5.3	52.5 ± 5.0 ^a^	58.6 ± 6.0 ^b,c,d^	50.7 ± 10.3	43.7 ± 6.3 ^b^	43.9 ± 5.8	48.3 ± 7.2 ^c^	36.9 ± 10.2 ^a,d^	<0.0001
cc	3.00 ± 1.42 ^a^	3.71 ± 1.36 ^b^	4.03 ± 1.41	4.55 ± 1.40	4.63 ± 2.74	6.55 ± 1.87 ^a,b^	4.02 ± 0.99	4.84 ± 1.7	0.0010
CTs	19.1 ± 6.1 ^a,b^	1.60 ± 0.5 ^a,c,d,e,f^	5.19 ± 4.12 ^b^	14.1 ± 6.62 ^c^	9.26 ± 5.79	12.5 ± 9.8 ^d^	11.9 ± 3.0 ^e^	21.2 ± 2.7 ^f^	<0.0001
ttt	17.2 ± 5.5 ^a,b^	0.72 ± 0.31 ^a,c,d,e,f^	4.79 ± 4.08 ^b^	11.8 ± 7.39 ^c^	6.07 ± 2.67	8.99 ± 8.61 ^d^	8.43 ± 2.96 ^e^	19.3 ± 3.2 ^f^	<0.0001
ttc/ctt	1.17 ± 1.00	0.33 ± 0.17 ^a^	0.70 ± 0.39	1.51 ± 1.58	1.47 ± 1.98	0.94 ± 0.60	1.23 ± 0.61 ^a^	0.81 ± 0.41	0.0162
cct	0.73 ± 0.58 ^a^	0.56 ± 0.28	0.24 ± 0.06 ^a,b,c,d^	0.65 ± 0.24 ^b^	0.74 ± 0.82	0.55 ± 0.17 ^c^	2.49 ± 1.61 ^d^	0.50 ± 0.21	<0.0001
(% CDs)									
tt	35.2 ± 5.8 ^a,b^	42.9 ± 4.6	35.9 ± 5.7	36.1 ± 8.9	45.2 ± 7.8 ^a^	41.9 ± 7.8	42.3 ± 5.1	47.1 ± 11.2 ^b^	0.0011
ct	61.0 ± 6.6 ^a,b,c^	53.3 ± 5.0	60.1 ± 5.7	58.6 ± 9.3	49.7 ± 6.7 ^a^	50.5 ± 7.4 ^b^	54.7 ± 7.1	46.8 ± 12.6 ^c^	0.0002
cc	3.72 ± 1.59 ^a^	3.77 ± 1.39 ^b^	4.15 ± 1.52 ^c^	5.32 ± 1.72	5.19 ± 2.93	7.56 ± 2.15 ^a,b,c^	4.58 ± 1.16	6.16 ± 2.22	0.0004
(% CTs)									
ttt	88.7 ± 9.3 ^a^	43.1 ± 13.9 ^a,b,c^	95.2 ± 72.7	78.1 ± 21.2 ^b^	70.4 ± 16.0	69.3 ± 23.9	70.5 ± 13.7	90.9 ± 9.0 ^c^	<0.0001
ttc/ctt	8.27 ± 8.06 ^a^	22.1 ± 9.6 ^a,b^	15.7 ± 5.2 ^c^	15.6 ± 18.8	15.6 ± 15.5	12.2 ± 8.9	10.0 ± 3.3	3.93 ± 2.43 ^b,c^	0.0002
cct	4.45 ± 4.35 ^a,b^	35.9 ± 16.4 ^a,c,d,e,f,g^	7.27 ± 6.04 ^c^	6.20 ± 4.35 ^d^	8.72 ± 8.88 ^e^	9.44 ± 8.35 ^f^	21.9 ± 14.9 ^b^	2.39 ± 0.90 ^g^	<0.0001

Data are shown as mean values ± standard deviation (SD). *p* value ≤ 0.05—significant differences among groups in Kruskal–Wallis test (*). Values sharing a letter in one row are significantly different (*p* < 0.05) in multiple comparison test; CFAs—conjugated fatty acids; CDs—conjugated dienes; CTs—conjugated trienes.

**Table 6 cancers-16-00479-t006:** Profile of FAs and CFAs in rats’ spleen in revealed clusters.

	Cl1	Cl2	Cl3	Cl4	*p* Value
C14:0	387.3 ± 117.9 ^a,b,c^	123.4 ± 45.8 ^a,d,e^	201.9 ± 105.2 ^b,d^	179.8 ± 64.9 ^c,e^	<0.0001
C15:0	153.2 ± 23.1 ^a,b,c^	67.79 ± 20.74 ^a,d,e^	91.87 ± 29.29 ^b,d^	95.88 ± 34.76 ^c,e^	<0.0001
C16:0	11,034 ± 1336 ^a,b^	3663 ± 931 ^a,c^	4581 ± 1049 ^b,d^	6038 ± 1100 ^c,d^	<0.0001
i-C17:0	187.5 ± 28.7 ^a,b^	71.83 ± 30.19 ^a,c^	88.25 ± 27.27 ^b^	111.5 ± 40.0 ^c^	<0.0001
C18:0	7579 ± 803 ^a,b^	2736 ± 676 ^a,c^	2951 ± 502 ^b,d^	4254 ± 825 ^c,d^	<0.0001
C20:0	74.32 ± 22.69 ^a,b,c^	28.49 ± 16.98 ^a^	33.50 ± 16.29 ^b^	36.03 ± 14.16 ^c^	0.0015
C22:0	68.80 ± 26.32 ^a,b^	20.23 ± 12.36 ^a,c^	30.71 ± 18.80 ^b^	36.41 ± 15.25 ^c^	<0.0001
ΣC16:1 ^#^	1056 ± 424 ^a,b,c^	252 ± 116 ^a,d^	456 ± 268 ^b^	399 ± 234 ^c,d^	0.0001
c9C17:1	108.1 ± 35.5 ^a,b,c^	42.65 ± 9.69 ^a^	46.03 ± 13.40 ^b^	56.60 ± 29.59 ^c^	0.0008
c9C18:1	6296 ± 1372 ^a,b^	1609 ± 555 ^a,c^	2232 ± 939 ^b^	2875 ± 1456 ^c^	<0.0001
c11C18:1	970.0 ± 136.4 ^a,b^	423.4 ± 94.7 ^a,c,d^	540.9 ± 130.0 ^b,c^	601.2 ± 168.2 ^d^	<0.0001
c11C20:1	128.3 ± 41.6 ^a,b^	59.41 ± 24.12 ^a,c^	72.46 ± 23.31 ^b^	79.40 ± 20.67 ^c^	0.0002
c9c12C18:2	15,332 ± 2582 ^a,b^	4269 ± 1439 ^a,c^	5321 ± 1595 ^b,d^	7560 ± 1735 ^c,d^	<0.0001
c9c12c15C18:3	463.3 ± 119.2 ^a,b,c^	94.89 ± 40.86 ^a^	86.81 ± 62.43 ^b,d^	163.5 ± 134.4 ^c,d^	<0.0001
c11c14C20:2	143.6 ± 26.0 ^a,b^	47.33 ± 17.80 ^a,c^	62.69 ± 19.40 ^b,d^	84.94 ± 25.94 ^c,d^	<0.0001
c8c11c14C20:3	201.8 ± 24.9 ^a,b^	81.53 ± 28.09 ^a,c^	100.1 ± 36.8 ^b,d^	158.8 ± 39.0 ^c,d^	<0.0001
c5c8c11c14C20:4	6687 ± 505 ^a,b^	3125 ± 634 ^a,c^	3229 ± 578 ^b,d^	4248 ± 933 ^c,d^	<0.0001
c5c8c11c14c17C20:5	134.9 ± 60.0 ^a^	73.02 ± 37.82	66.48 ± 57.70 ^a,b^	121.3 ± 92.2 ^b^	0.0027
c7c10c13c16c19C22:5	829.3 ± 111.8 ^a,b^	253.6 ± 95.4 ^a,c^	222.2 ± 98.2 ^b,d^	549.4 ± 161.5 ^c,d^	<0.0001
c4c7c10c13c16c19C22:6	544.8 ± 95.8 ^a,b^	239.5 ± 78.1 ^a,c^	256.2 ± 66.1 ^b,d^	380.2 ± 86.3 ^c,d^	<0.0001
Sum FA	52,637 ± 5506 ^a,b^	17,264 ± 4245 ^a,c^	20,827 ± 4594 ^b,d^	28,351 ± 4855 ^c,d^	<0.0001
CFA	755.5 ± 222.4	563.7 ± 154.9 ^a^	855.4 ± 186.1 ^a,b^	516.6 ± 129.0 ^b^	<0.0001
CD	670.3 ± 181.4	492.8 ± 130.0 ^a^	745.2 ± 142.6 ^a,b^	488.1 ± 112.8 ^b^	<0.0001
CT	85.23 ± 47.43 ^a^	61.11 ± 40.68 ^b^	118.9 ± 85.2 ^c^	28.53 ± 44.16 ^a,b,c^	<0.0001
tt	199.2 ± 22.4 ^a^	216.3 ± 50.8 ^b^	318.1 ± 89.4 ^a,b,c^	195.6 ± 43.3 ^c^	<0.0001
ct	433.8 ± 155.9	249.5 ± 88.9 ^a^	390.4 ± 109.1 ^a,b^	271.7 ± 86.6 ^b^	<0.0001
cc	37.28 ± 15.68	26.99 ± 13.34 ^a^	40.66 ± 18.33 ^a,b^	20.82 ± 9.34 ^b^	<0.0001
ttt	66.32 ± 44.08	50.11 ± 37.84 ^a^	83.23 ± 64.06 ^b^	23.72 ± 41.99 ^a,b^	<0.0001
ttc/ctt	13.12 ± 13.47	6.43 ± 8.53 ^a^	8.52 ± 5.85 ^b^	3.17 ± 4.46 ^a,b^	0.0001
cct	5.79 ± 3.24	3.74 ± 3.85 ^a^	9.75 ± 12.54 ^a,b^	2.48 ± 1.53 ^b^	0.0001

Data are shown as mean values ± standard deviation (SD). *p* value ≤ 0.05—significant differences among groups in Kruskal–Wallis test. Values sharing a letter in one row are significantly different (*p* < 0.05) in multiple comparison test; CFAs—conjugated fatty acids; CDs—conjugated dienes; CTs—conjugated trienes; ΣC16:1 ^#^—sum of two C16:1 isomers detected in spleen samples was presented, due to inability of their exact identification.

## Data Availability

Data are contained within the article and Appendix A.

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
