# Peer review of "Exploring the Influence of the Selected Conjugated Fatty Acids Isomers and Cancerous Process on the Fatty Acids Profile of Spleen"

_cancers, 2024, doi:10.3390/cancers16030479_

Round 1

Reviewer 1 Report

Comments and Suggestions for Authors

This study evaluated the effects of diet supplementary with pomegranate seed oil and bitter melon extract on concentrations of fatty acids (FAs) in the rat’s spleen in the context of carcinogenesis and found that dietary supplementation leads to alterations in the spleen's FA profile and affect the cancerous process. Intake of unsaturated fatty acids has long been thought to be beneficial to health. But how it benefits health is still unknown. Therefore, the author's research is very meaningful. However, some clarification and improvements should be made before acceptance, especially the methods and results in this study. 

1. The author emphasizes the use of advanced technology “gas chromatography-mass spectrometry (GC-MS) and silver ion- ion-impregnated high-performance liquid chromatography with photodiode array detection (Ag+-HPLC-DAD)” many times in the article. However, the advancement of technology is relative and open. Technology is only a tool for revealing scientific problems. Unless the technique is original by the author, it should not be overemphasized. I think readers are more concerned about what scientific problems the author reveals or solves than what tools the author uses. 

2. The writing in this paper is rigmarole, especially the introduction and discussion. It is suggested that the author systematically analyze and elaborate the key results of this study.

3. I don't think the animal experiment in the study is reasonable. The authors said that the diet supplement was administered via gavage. So, did the authors consider the effects of the brain-gut axis or entero-liver axis on animal metabolism? How did the authors decide on the gavage dose? Do these products have a dose effect? Does gavage affect the diet of experimental animals? How did the food and water intake, body weight, serum metabolites, visceral index, and other physical indicators of the animals in each group change? The authors said, “After the experimental period all animals from each group were decapitated and exsanguinated”. So, does this method of killing animals comply with the code of ethics for laboratory animals? I don't think this is an appropriate approach either for the laboratory animals or for sample collection. 

4. Lines 116 and 129: The authors should provide a brief description of the information from the reference, so readers can comprehend your analysis without checking another document. 

5. The statistical analysis of the data in the tables is confusing for me, especially the significance analysis. I can't understand the author's analysis well through the tables.

6. Without the data about the feed formula, food and water intake, body weight, visceral index, and other physical indicators of the animals, I think it is impossible to speculate on the effects of dietary supplementation on animal metabolism, especially the lipid metabolism which closely related to animal energy metabolism. 

7. The authors studied the relationship between the fatty acid composition of the spleen and cancer. However, it is well known that the liver is the most important organ for fat synthesis and metabolism. In this study, how do the authors consider the liver's impact on cancer? Or whose fat composition has a greater impact on cancer, the liver, or the spleen (or other tissues and organs)? How does the author's method apply to clinical testing? Like how can you harvest human spleen tissue? Or how to detect the lipid composition of the human spleen in a non-invasive or minimally invasive manner? 

8. As one of the most important immune organs, the authors should design experiments to further examine the effects of fat composition on inflammation and innate and adaptive immune response. The authors should design and supply more tests, the current results are not enough to support the conclusions of the paper.

Author Response

Reviewer 1

Comments and Suggestions for Authors

This study evaluated the effects of diet supplementary with pomegranate seed oil and bitter melon extract on concentrations of fatty acids (FAs) in the rat’s spleen in the context of carcinogenesis and found that dietary supplementation leads to alterations in the spleen's FA profile and affect the cancerous process. Intake of unsaturated fatty acids has long been thought to be beneficial to health. But how it benefits health is still unknown. Therefore, the author's research is very meaningful. However, some clarification and improvements should be made before acceptance, especially the methods and results in this study. 

We are greatly obliged for having received the Reviewer’s 1 opinions and comments on our manuscript. We are very grateful for the time and effort they spared for revision of our article. We have read thoroughly all valuable comments, advice and suggestions and we found them really inspiring and helpful. The replies to specific comments are listed below. All changes in the manuscript are marked in navy blue. We do hope you will find them suitable and sufficient.

  1. The author emphasizes the use of advanced technology “gas chromatography-mass spectrometry (GC-MS) and silver ion- ion-impregnated high-performance liquid chromatography with photodiode array detection (Ag+-HPLC-DAD)” many times in the article. However, the advancement of technology is relative and open. Technology is only a tool for revealing scientific problems. Unless the technique is original by the author, it should not be overemphasized. I think readers are more concerned about what scientific problems the author reveals or solves than what tools the author uses. 

Response: We are very grateful for this remark. We agree that analytical technique is essential for solving scientific problem, especially in context of choosing the proper technique, tailored to the problem being investigated/solved. We have often emphasize the meaning of combining of our original GC-MS method (BiaÅ‚ek et al. Anim Feed Sci Technol 2018, 241, 63–74, doi:10.1016/j.anifeedsci.2018.04.015) with our original Ag+-HPLC-DAD method (Argentometric Liquid-Chromatography, Czauderna et al. Acta Chromatogr 2007, 59–71) for reviling the total fatty acids profile, especially for the scientists interested in conjugated fatty acids analysis. In our paper concerning pomegranate seed oil analysis we presented the detailed explanation of this issue (BiaÅ‚ek et al. Journal of Dietary Supplements 2021; 18: 4, 351-371; doi.org/10.1080/19390211.2020.1770394). As we stated there, it is impossible to assess the total profile of conjugated fatty acids solely with GC-MS and the combination of this technique with Ag+-HPLC-DAD gives more detailed insight into the isomers of conjugated fatty acids present in a sample. Importantly, compared to GC-MS methods, our original Ag+-HPLC-DAD method using four columns loaded with Ag+ and photodiode detection, allows selective and specific detection (in the UV-range from 230 to 300 nm) of conjugated fatty acids (CFAs); the remaining non-conjugated fatty acids are undetected in this wavelength range (i.e. 230 - 300 nm). Therefore, the argentometric method should be used to determine CFAs (even up to 60 CFAs simultaneously) (BiaÅ‚ek et al., J. Diet. Suppl.  2021; 18:4, 351-371). In the context of differences in physiological and pathological conditions coupled with supplementation of CFAs dietary sources in our described experiment it seems of utmost importance. As far as we are aware, such attempt is not very common (to our best knowledge only Professor Sebastiano Banni and his team from University of Cagliari and our team are using this technique quite often) but is very resultful. Thus, we want to emphasize such approach, as one of the novel aspect of our research. However, according to Reviewer’ suggestion we implemented more detailed explanation of our analytical and scientific attempt (lines 97-101).

  1. The writing in this paper is rigmarole, especially the introduction and discussion. It is suggested that the author systematically analyze and elaborate the key results of this study.

Response: As suggested by the Reviewer 1 proper changes has been introduced into our manuscript.

  1. I don't think the animal experiment in the study is reasonable. The authors said that the diet supplement was administered via gavage. So, did the authors consider the effects of the brain-gut axis or entero-liver axis on animal metabolism? How did the authors decide on the gavage dose? Do these products have a dose effect? Does gavage affect the diet of experimental animals? How did the food and water intake, body weight, serum metabolites, visceral index, and other physical indicators of the animals in each group change? The authors said, “After the experimental period all animals from each group were decapitated and exsanguinated”. So, does this method of killing animals comply with the code of ethics for laboratory animals? I don't think this is an appropriate approach either for the laboratory animals or for sample collection. 

Response: We are grateful for all these questions from Reviewer 1 which we are glad to answer. In case of examination of the influence of dietary ingredients’ in animal experiments two-pronged approach can be observed: (i) indirect, where examined ingredient is added into the fodder, and it is difficult to certainly exam the amount of consumed supplement or (ii) direct, where supplement is administrated into the selected part of digestive tract and we are sure that total amount of administrated ingredient will be consumed. Experiment design must comply with the physico-chemical properties of the ingredient, its activity (which corresponds to the dose) and the main objective of the experiment. Presented experiment was a continuation of our previous studies performed in the same breast cancer animal model with conjugated linoleic acid, where we used two doses (1% or 2% of CLA in diet of animals – assessed based on data from scientific papers referring to biologically active share of CLA, which equals to 0.15 ml or 0.30 ml of dietary supplement in form of oil). In present experiment we assumed that PSO as an oil rich in punicic acid (one of CLnA isomers), to be an indirect source of CLA. We adjusted the dose of the oil (0.15 ml per day) to our earlier experiments. As previously we have decided to use the gavage to be able to introduce exact, fixed dose of the supplement, to examine the dose-effect dependance and to be able to compare obtained results to our previous results with CLA. We were also convinced that 0.15 ml is an amount, which may be easily and safely given to animals. Usage of intragastrical administration via gavage, which size is adjusted to the size of an animal, is approved in animal experiments and was used in numerous experiments on rat model by our team. It was done by very experienced technician, with great experience in this procedure, with whom animals were accustomed through handling, which reduced their distress to minimum. It does not affect the regular feed intake, growth parameters and overall condition of the animals, which was proved numerous times in our previous animal experiments. Moreover, it should be emphasized, that the entire experiment design, including usage of the gavage was approved by the 2nd Local Ethical Committee on Animal Experiments consent (No. 56/2013 and 54/2015) in accordance with European Union Directive 2010/63/EU for animal experiments, which was mentioned in the ‘Materials and methods’ section of our manuscript (lines 129-131). Also decapitation as a proper way of animals’ euthanasia was approved as the part of the experiment by 2nd Local Ethical Committee on Animal Experiments consent (No. 56/2013 and 54/2015) in accordance with European Union Directive 2010/63/EU for animal experiments (see annex IV to this Directive). There were no differences in feed and liquid intake between experimental groups, which was checked/controlled during the entire experiment. Similarly as main growth parameters these data were presented in our previous papers (Lepionka et al. Prostaglandins Other Lipid Mediat 2019, 142, 33–45, doi:10.1016/j.prostaglandins.2019.03.005). To avoid the plagiarism we did not present these results in this manuscript, however, we mentioned it in the text with proper references (lines 162-155 and 623-627). In the present experiment the effects of the brain-gut axis or entero-liver axis on animal metabolism were not considered. However we are very grateful of Reviewer 1 for this suggestion, as we find it very inspiring for our future research. We do hope that this explanations are informative and sufficient for the Reviewer 1.

  1. Lines 116 and 129: The authors should provide a brief description of the information from the reference, so readers can comprehend your analysis without checking another document. 

Response: As mentioned before to avoid plagiarism we decided not to repeat the results which were previously published (Lepionka et al. Prostaglandins Other Lipid Mediat 2019, 142, 33–45, doi:10.1016/j.prostaglandins.2019.03.005). However, due to this suggestion of Reviewer 1, we have included short summary of previous results (see lines 120-127 and 140-142 of revised version).

  1. The statistical analysis of the data in the tables is confusing for me, especially the significance analysis. I can't understand the author's analysis well through the tables.

Response: We would like to explain that all data were presented as mean values ± standard deviation. For variables with skew distribution, data were transformed into logarithms, retransformed after calculations and presented as mean and confidence interval. Statistica 13.5 software (StatSoft, Cracow, Poland) was used for the statistical analysis. For variables with normal distribution obtained data were tested with one-way ANOVA and post hoc Tuckey test (marked * in tables). For variables without normal distribution data were tested with the Kruskal–Wallis test, which is a non-parametric equivalent of one-way ANOVA, with post hoc Dunn’s test. For values of indices calculated based on FA content the Kruskal–Wallis test, with post hoc Dunn’s test were performed. The acceptable level of significance was established at p < 0.05. Values sharing a letter in one, singular row were significantly different on one of the above mentioned post hoc test. We would like to assure that this statistical approach was used and approved in numerous of our previously published papers describing similar results. Please refer to our earlier papers, e.g.. Lepionka et al. Prostaglandins Other Lipid Mediat 2019, 142, 33–45; doi:10.1016/j.prostaglandins.2019.03.005, BiaÅ‚ek et al. Antioxidants 2020, 9, 243; doi:10.3390/antiox9030243, BiaÅ‚ek et al. Molecules 2020, 25, 1814; doi:10.3390/molecules25081814, Lepionka et al. Prostaglandins and Other Lipid Mediators 2021; 152: 106495; doi.org/10.1016/j.prostaglandins.2020.106495 or BiaÅ‚ek et al. Chemistry and Physics of Lipids 2021; 235: 105057 10.1016/j.chemphyslip.2021.105057

  1. Without the data about the feed formula, food and water intake, body weight, visceral index, and other physical indicators of the animals, I think it is impossible to speculate on the effects of dietary supplementation on animal metabolism, especially the lipid metabolism which closely related to animal energy metabolism. 

Response: As it was mentioned before these parameters concerning feed and liquid intake as well as growth performance parameters were previously published and to avoid plagiarism cannot be presented in this manuscript. They were only mentioned in the ‘Discussion’ section with proper references. However, due to this remark of Reviewer 1, we included additional explanation in the “Materials and methods’ section of our manuscript (see lines 162-155 and 623-627).

  1. The authors studied the relationship between the fatty acid composition of the spleen and cancer. However, it is well known that the liver is the most important organ for fat synthesis and metabolism. In this study, how do the authors consider the liver's impact on cancer? Or whose fat composition has a greater impact on cancer, the liver, or the spleen (or other tissues and organs)? How does the author's method apply to clinical testing? Like how can you harvest human spleen tissue? Or how to detect the lipid composition of the human spleen in a non-invasive or minimally invasive manner? 

Response: We would like to explain, that fatty acids profile in livers as well as in hepatic microsomes in this model experiment were investigated and published previously: (i) Lepionka T, Białek M, Czauderna M, Białek A. Pomegranate seed oil and bitter melon extract supplemented in diet influence the lipid profile and intensity of peroxidation in livers of SPRD rats exposed to a chemical carcinogen. Prostaglandins and Other Lipid Mediators 2021; 152: 106495; doi.org/10.1016/j.prostaglandins.2020.106495 and (ii) Lepionka T, Białek M, Czauderna M, Szlis M, Białek A. Lipidomic profile and enzymes activity in hepatic microsomes of rats in physiological and pathological conditions. International Journal of Molecular Sciences 2022; 23: 442; doi.org/10.3390/ijms23010442.
We also investigated the lipidomic profile of mammary tumors (Białek A, Jelińska M, Białek M, Lepionka T, Czerwonka M, Czauderna M. The Effect of diet supplementation with pomegranate and bitter melon on lipidomic profile of serum and cancerous tissues of rats with mammary tumours. Antioxidants 2020; 9: 243; doi:10.3390/antiox9030243) and cardiomyocytes (Białek A, Białek M, Lepionka T, Pachniewicz P, Czauderna M. Oxysterols and lipidomic profile of myocardium of rats supplemented with pomegranate seed oil and/or bitter melon aqueous extract - cardio-oncological animal model research. Chemistry and Physics of Lipids 2021; 235: 105057 10.1016/j.chemphyslip.2021.105057). Spleen has been recognized lately as the very important organ in lipid metabolism thus we decided to extend our previous studies on PSO and BME to include also studies in spleen to investigate the influence of applied supplementation and co-existing cancerous process on FA and CFA profile in this organ. FA and CFA profile of spleen, liver and also other organs result both from applied dietary supplementation and pathological process (mammary tumors). Our studies show that cancerous process to the greater extend affects these parameters than PSO and BME. Regarding harvesting of human spleen tissue biopsy can be performed to obtain fragment for lipid investigation, which may be considered as minimally invasive procedure. We are grateful for these remarks as we have found them extremely inspiring for our future research in collaboration with clinicians.

  1. As one of the most important immune organs, the authors should design experiments to further examine the effects of fat composition on inflammation and innate and adaptive immune response. The authors should design and supply more tests, the current results are not enough to support the conclusions of the paper.

Response: We are very grateful for this remark, we find it very inspiring and for sure we will take it into consideration when preparing our next experiments. As suggested, the conclusion was rewritten to be supported by the obtained results.

Reviewer 2 Report

Comments and Suggestions for Authors

The authors performed an animal experiment to seek the associations between fatty acids and cancer in the spleen, and most of the work presented was FA analysis. The issues below should be noticed:

1. The authors detected many FAME isomers, showing very good resolution of GC. Please the authors provide the FAME separation chromatography of the real sample, together with the standards, as supplementary files.

2. Table 1. Is it a quantitative experiment or a semi-quantitative experiment? Please make it clear.

3. Line 208. The authors used many indices, but please claim their significance and at least, show the reasons and references.

4. HCA did not provide sufficient quantitative findings, which just described a trend of the FAME results. What do the authors think about it? I suggest that the authors use some quantitative data for their observations and findings.

5. Were the detected FA endogenous (except for the fed CFAs)?

6. Lipid metabolism cannot be elucidated by only FA profiling. Actually, it is a quite complex system. Without biochemical assays, the authors are not recommended to do too much deduction. So, please change the related descriptions and demonstrations.

Comments on the Quality of English Language

Moderate editing of English language required.

Author Response

Reviewer 2

We are greatly obliged for having received the Reviewer’s 2 opinions and comments on our manuscript. We are very grateful for the time and effort they spared for revision of our article. We have read thoroughly all valuable comments, advice and suggestions and we found them really inspiring and helpful. The replies to specific comments are listed below. All changes in the manuscript are marked in green. We do hope you will find them suitable and sufficient.

The authors performed an animal experiment to seek the associations between fatty acids and cancer in the spleen, and most of the work presented was FA analysis. The issues below should be noticed:

  1. The authors detected many FAME isomers, showing very good resolution of GC. Please the authors provide the FAME separation chromatography of the real sample, together with the standards, as supplementary files.

Response: We would like to thank Reviewer 2 for this remark. The good GC resolution in our research was obtained mainly due to usage of very long capillary column (120 m) as well as the shallow temperature gradient refined in details. As suggested an exemplary chromatogram has been included as Figure 1 and chromatogram of FAME standards has been included in supplementary materials (Figure S1).

  1. Table 1. Is it a quantitative experiment or a semi-quantitative experiment? Please make it clear.

Response: We would like to assure that it is quantitative experiment. It was mentioned in the ‘Introduction’ where the aim of the study was explained (line 91). Results were expressed both per mass of fresh tissue and then calculated into percentage share in the total FA pool. Similarly, CFAs were expressed both per mass of a fresh tissue and then calculated into percentage share in total CFAs as well as in CD and CT. These both ways of presentation of results were used in our previously published papers, e.g.: BiaÅ‚ek et al. Antioxidants 2020; 9: 243; doi:10.3390/antiox9030243, Lepionka et al. Prostaglandins and Other Lipid Mediators 2021; 152: 106495; doi.org/10.1016/j.prostaglandins.2020.106495, or BiaÅ‚ek et al. Chemistry and Physics of Lipids 2021; 235: 105057 10.1016/j.chemphyslip.2021.105057. Proper explanation has been given in the ‘Materials and methods’ section (lines 202-203 and 233-234) and included into the footnotes of Table 1.

  1. Line 208. The authors used many indices, but please claim their significance and at least, show the reasons and references.

Response: Indices, which were used in our manuscript, were developed, published and successfully used by other authors, e.g. Ulbricht et al. Lancet 1991; 338: 985-992, Moran et al. Food Chemistry 2013; 138: 2407-2414, Fernandez et al. Food Chemistry 2007; 101: 107-112. We have decided to include indices calculated on the basis of particular determination in our manuscript, because of their similar bioactive properties (e.g. sum of atherogenic and thrombogenic fatty acids, i.e. A-SFA and T-SFA) as well as due to the fact that the ultimate effect of these compounds in the organism does not simply depend on their content but is a resultant of their mutual interplay (e.g. indexASFA, indexTSFA, h/H-Ch). We believe that calculation of indices is the easiest way to show some dependencies among determined compounds and such approach may draw readers' attention to the existence of such dependencies and encourage them to delve deeper into the topic. Moreover, not only the fatty acids content in the diet influence their effects in the body, but also their endogenous metabolism of fatty acids, e.g. by elongation and desaturation. That is why we have decided to calculate the relationships (ratios) between fatty acid products and precursors to estimate the enzyme activities. We are aware that these ratios cannot be assumed to directly reflect enzymes activities, but approach was previously used as an indirect way to measure enzymes activities, e.g. Stawarska et al. Prostaglandins, Leukotrienes and Essential Fatty Acids (PLEFA) 2018; 128: 62-66 or Stawarska et al. Molecules 2020, 25, 5232; doi:10.3390/molecules25225232. Desaturases are widely expressed in many tissues and their expression is regulated by various factors, e.g. different nutrients including fatty acids. Polymorphisms in FADS1 and FADS2 genes, encoding D5D and Δ6-desaturase (D6D), are associated with the ratios of arachidonic acid (C20:4 n-6) to linoleic acid (C18:2 n-6) and eicosapentaenoic acid (C20:5 n-3) to α-linolenic acid (C18:3 n-3). Additionally, PUFA are known to influence desaturase activities. In our previous research we have confirmed that CLA affects the metabolism of LA, by competing for Δ5- and Δ6-desaturases in livers (BiaÅ‚ek et al. Acta Poloniae Pharmaceutica - Drug Research, Vol. 71 No. 5 pp. 747-761, 2014; ), which leads to decreased levels of LA metabolites. It is possible that CLnA present in G, GM, Gplus and GMplus diets also may compete with LA and ALA and reduce their availability for enzymes converting them to non-conjugated FA. The calculated indices are measurable informations (the final ‘numerical’ conclusions) describing the impact of fatty acid concentrations in the spleen on selected physiological processes and etiologies of the diseases discussed. Taking all these premises into account we have decided to include indices in our manuscript. Similar approach has been used in our recently published paper regarding the influence of organic and inorganic selenium compounds on lipidomic profile of spleen of lambs (BiaÅ‚ek et al. Animals. 2024; 14: 133. https://doi.org/10.3390/ani14010133). We do hope that Reviewer 2 will find this explanation sufficient.

  1. HCA did not provide sufficient quantitative findings, which just described a trend of the FAME results. What do the authors think about it? I suggest that the authors use some quantitative data for their observations and findings.

Response: All data presented in the tables 1-5 are quantitative, which was mentioned in the aim of the study and described in ‘Materials and methods’ section. Obtained quantitative data were subjected to statistical analysis and described in results section. CA was performed based on these quantitative data. We do agree that it shows the similarity of FA profiles in individuals. However, based on revealed similarities, FA and CFA content in revealed clusters was calculated and statistical analysis (Kruskal-Wallis test with multiple comparison test) was performed. These results are given in Table 6. We would like to emphasize, that CA was performed as additional, more advanced analysis simply to verify if any new trends and dependances among investigated variables will be revealed.

  1. Were the detected FA endogenous (except for the fed CFAs)?

Response: We would like to explain, that animal feed given to animals (Labofeed H) contained fat and the fatty acids profile of it was published previously (Lepionka et al. Prostaglandins Other Lipid Mediat 2019, 142, 33–45, doi:10.1016/j.prostaglandins.2019.03.005.). Moreover, PSO also contained numerous fatty acids, apart CFA (data previously published – Lepionka et al. Prostaglandins Other Lipid Mediat 2019, 142, 33–45, doi:10.1016/j.prostaglandins.2019.03.005). Fatty acids profile in spleen results from the dietary intake and endogenous fatty acids metabolism, which is multistep, complex process depending on enzymes activity, interaction among fatty acids and other compounds, genes and proteins expression etc. Moreover, the concentrations of fatty acids in the spleen are the result of anabolic and catabolic lipid processes. As we did not use any labeling of fatty acids providing with diet, we assume, that the detected fatty acids profile results both from dietary intake and endogenous synthesis. Indices of desaturase and elongase activities may give a slight glimpse into efficiency of endogenous synthesis.

  1. Lipid metabolism cannot be elucidated by only FA profiling. Actually, it is a quite complex system. Without biochemical assays, the authors are not recommended to do too much deduction. So, please change the related descriptions and demonstrations.

Response: We would like to thank Reviewer 2 for this remark. We are aware of the complexity of lipid metabolism in the organism as lipidomics is one of the main areas of our scientific interests. As CFAs were expected as main active dietary factors in this experiment, we decided to focus on the fatty acids profiling with the special emphasis on CFA isomers content, which is according to our knowledge, the innovative approach. As this research is our first attempt to evaluate the influence of dietary supplementation and cancerous process on spleen, we tried to focus on the most commonly investigated parameters, i.e. fatty acids. However, we find remark about biochemical assays very inspiring and we will take them into account when we will plan our future experiments. According to Reviewer’ suggestion, some modifications have been introduced, especially in this part of manuscript, which might have been considered as too far-reaching.

Reviewer 3 Report

Comments and Suggestions for Authors

 Dear Editor

the experimental part related to the work up of the samples is not reported.

Further it's not clear  how the unambigously determination of positional and geometrical isomers of  MUFA fatty acid is performed ( MUFA 16:1 and 18:1)

So provide this detailed, GC chromatograms and Enlargments of GC chromatograms in order that it's possible to identify each peak otherwise the results are not correct

Comments on the Quality of English Language

no comments

Author Response

Reviewer 3

We are greatly obliged for having received the Reviewer’s 3 opinions and comments on our manuscript. We are very grateful for the time and effort they spared for revision of our article. We have read thoroughly all valuable comments, advice and suggestions and we found them really inspiring and helpful. The replies to specific comments are listed below. All changes in the manuscript are marked in brown. We do hope you will find them suitable and sufficient.

Dear Editor

the experimental part related to the work up of the samples is not reported.

Response: We would like to explain, that some original data concerning the experiment, has been published previously and we decided not to repeat them due to the plagiarism. Only proper references has been introduced in the initial version of the manuscript (BiaÅ‚ek et al. Anim Feed Sci Technol 2018, 241, 63–74, doi:10.1016/j.anifeedsci.2018.04.015, Czauderna et al. Acta Chromatogr 2007, 59–71 and Czauderna et al. Czech Journal of Animal Science 2011, 56, 23–29). However, taking into account this remark, more detailed description has been introduced into the manuscript (lines 169-184).

Further it's not clear  how the unambigously determination of positional and geometrical isomers of  MUFA fatty acid is performed ( MUFA 16:1 and 18:1)

Response: We would like to explain that exemplary chromatogram of experimental sample has been introduced into the manuscript (Figure 1), according to recommendation of Reviewer 1. Moreover, FAME standards chromatogram has been introduced into Supplementary Materials (Figure S1). Moreover, we would like to emphasize that, as mentioned in the ‘Materials and methods’ section, FAME identification was based on electron impact ionization spectra of FAME and compared to authentic FAME standards, e.g. Supelco 37 Component FAME Mix and the NIST 2007 reference mass spectra library (National Institute of Standard and Technology, Gaithersburg, MD, USA). Hence, all FAME analyses were based on total ion current chromatograms (TIC) and/or selected-ion monitoring chromatograms (SIM). If there were any doubts concerning the identification of C16:1 and C18:1 isomers, samples were spiked with single standards (if available) or they were identified based on retention times, the fixed elution order of fatty acids (order is determined by the boiling point of individual fatty acids which is connected with the carbon chain length (the longer the chain, the higher the boiling point), presence and number of double bonds (SFA, MUFA and PUFA of the same chain length), their localization in chain (the closer to carboxyl group, the lower boiling point) as well as their conformation (trans isomers proceeding cis isomers) and data from the other papers.

Moreover, we verified the elution order of the detected positional and geometric isomers C16:1 and C18:1 in the spleen with GC-chromatograms published in prominent analytical journals (e.g.: Kramer et al. Lipids 2008; 43:259–273, Stolyhwo et al. Food Anal. Methods (2013) 6:457–469).

So provide this detailed, GC chromatograms and Enlargments of GC chromatograms in order that it's possible to identify each peak otherwise the results are not correct

Response: As suggested by the Reviewer, the exemplary chromatogram has been introduced into the manuscript (Figure 1) and its enlarged fragment has been given below.

L.p.

Retention time [min]

Fatty acid name

1

58.049

t9 C16:1

2

58.663

c7 C16: 1

3

61.402

c9 C16:1

4

63.683

C17:0

5

67.110

c9 C17:1

6

71.938

C18:0

7

74.277

c6 C18:1

8

74.778

c7 C18:1

9

75.048

c9 C18:1

10

76.153

c11 C18:1

Round 2

Reviewer 1 Report

Comments and Suggestions for Authors

Although some problems still exist, the author answered all of the questions carefully. I am partially satisfied with the response to my concerns.

Author Response

Reviewer #1:

Although some problems still exist, the author answered all of the questions carefully. I am partially satisfied with the response to my concerns.

Response: We would like to thank the Reviewer 1 for his time and effort on reviewing our manuscript as well as for appreciation of corrections and improvements we have made.  

Reviewer 2 Report

Comments and Suggestions for Authors

The authors made proper modifications and improved the quality of the manuscript, especially the details of GC analysis. Still, I would like to point out that this was a semi-quantitative work because there were no calibration curves, no methodology validation, and no matrix effect assay. The resulting FA data were actually "estimated using peak area" and "calculated using standards". So, please the authors notice this issue.

Comments on the Quality of English Language

The English language has been improved.

Author Response

Reviewer #2:

The authors made proper modifications and improved the quality of the manuscript, especially the details of GC analysis. Still, I would like to point out that this was a semi-quantitative work because there were no calibration curves, no methodology validation, and no matrix effect assay. The resulting FA data were actually "estimated using peak area" and "calculated using standards". So, please the authors notice this issue.

Response: We are greatly obliged to Reviewer #2 for noticing our effort in improving the quality of our manuscript. However we cannot agree with Reviewer’s #2 statement, that our work is semi-quantitative. We would like to emphasize, that for quantification of a content of each assayed fatty acid (as FAME) a linear calibration equation (in the range of fatty acid concentrations expected in the assayed biological samples) was used, as required by good chromatographic practice. However, on specific request of Reviewer #2 we may deliver full set of these linear calibration equations (or please refer to our previously published paper: Journal of Animal and Feed Sciences, 2023, 32, 4, 385–399). Moreover, the whole pre-column method and GC-MS procedure were validated and re-validated, and have been successfully used in investigation of different tissues from numerous experiments on humans (e.g. pilots, sport players), experimental animals (e.g. rats, sheep, pigs, rabbits) as well as food products (e.g. ripening cheeses, vegetable oils) and animal products (e.g.: cow and ovine milk, fish oils as well as hen eggs and meat). The results of those experiments, including quantitative determination of fatty acids content, were published in numerous scientific journals of international scope and high impact factor. These papers are available for global audience. Hence we would like to emphasize, that in case of our pre-column method of sample preparation, the matrix effect is infinitesimal/negligible, because: (i) for saponification of animal and plant tissues we used methanolic solution of potassium hydroxide (KOH); (ii) released free fatty acids were extracted with organic solvents: dichloromethane and hexane (or heptane); (iii) after methylation (derivatization) of free fatty acids, formed FAMEs were extracted with GC-hexane of ≥99% purity (Acta Chromatographica 2007, 18, 59 – 71). During all these steps the matrix  (containing proteins, carbohydrates, etc.) from assayed biological samples was removed and resulted transparent solution was injected into GC-MS. Furthermore, we used the long capillary column (120 m) and the very shallow gradient column temperature program; therefore, trace amounts of some endogenous species of assayed animal and plant tissues do not interfere with analytical FAME peaks. This is evidenced by FAME-peak purity (i.e. “MS-similarity”) and particularly by excellent FAME-peak symmetry.

We do hope that Reviewer #2 will find our explanation sufficient.

Reviewer 3 Report

Comments and Suggestions for Authors

Dear Authors the  perfect identification of fatty acid is crucial for this kind of research.

 In the table 2  you showed the list of fatty acids. It's very strange for me that you identified the trans 9 16:1 and not 9 cis 16:1 (  that is totally absent in the sample); instead you identified 7cis 16:1;

it's not sufficient to use the ritention time of the fatty acids because you could have the overlap

for example 16:1 can have 3 different typs of position isomers ( 6 cis, 9 cis and 7 cis  derived from 3 different metabolic pathways) ; ( new  articles and reviewes about this topic are published) ; of course  for each   positional isomers there is the corrisponding geometrical isomers; the positional isomerism is crucial in the cancer

There  are  synthetical  and analytical procedures able to distinguish the positional isomers based  diagnostic fragmentation after derivatization

The work is not completed

Comments on the Quality of English Language

No comment

Author Response

Reviewer #3:

Dear Authors the perfect identification of fatty acid is crucial for this kind of research.

 In the table 2 you showed the list of fatty acids. It's very strange for me that you identified the trans 9 16:1 and not 9 cis 16:1 (that is totally absent in the sample); instead you identified 7cis 16:1;

It's not sufficient to use the ritention time of the fatty acids because you could have the overlap

for example 16:1 can have 3 different typs of position isomers ( 6 cis, 9 cis and 7 cis  derived from 3 different metabolic pathways) ; ( new  articles and reviewes about this topic are published) ; of course  for each   positional isomers there is the corrisponding geometrical isomers; the positional isomerism is crucial in the cancer

There are synthetical and analytical procedures able to distinguish the positional isomers based  diagnostic fragmentation after derivatization

The work is not completed

Response: We totally agree with Reviewer #3 that based on retention time (RT) solely satisfactory identification of FA isomers cannot be done. That is why we used not only RT parameter but also MS spectra, the fixed elution order, spiking of the samples with standards and comparison with data from papers of other Authors, as it was mentioned in our previous response. We would like to emphasize that fatty acids identification in spleens was also made in comparison to data obtained for other tissues acquired in this experiment. As an example, in chromatograms of spleen we observed different peak pattern than in the liver. That is why it seems justified for us that also C16:1 isomeric composition of spleen will be different than of livers, where only cis9C16:1 isomer was detected (Lepionka et al. 2021, https://doi.org/10.1016/j.prostaglandins.2020.106495). We do agree that to be one hundred percent certain of identification of positional and geometric isomers of fatty acids, preparation of derivatives other than FAME (e.g. DMOX or DMDS) is necessary. However, preparation of such compounds is impossible in our case. We do not have any spleens left in our disposal, as the remaining parts of tissues were subjected to ICP-MS analysis for determination of minerals content. Due to the concerns of Reviewer#3, which we are unable to address with full certainty, we have decided to count a “Sum of C16:1 isomers- ΣC16:1” instead of presenting amounts of each compound separately. Due to this fact totally new statistical and chemometric analyzes were performed and an appropriate changes (marked in blue) were made in Tables 1 and 2 as well as in the manuscript body (cluster analysis). Moreover, new figures (Figures 1-3) and new table 6 have been prepared. We do hope that Reviewer #3 will find such approach sufficient and kindly accept these amendments.

Round 3

Reviewer 3 Report

Comments and Suggestions for Authors

no additional comments

Comments on the Quality of English Language

No comments